

# Source, transport and fate of soil organic matter inferred from microbial biomarker lipids on the East Siberian Arctic Shelf

Juliane Bischoff[1,*], Robert B. Sparkes[2,**], Ayça Doğrul Selver[2,3], Robert G. M. Spencer[4],
Örjan Gustafsson[5], Igor P. Semiletov[6,7,8], Oleg V. Dudarev[7,8], Dirk Wagner[9], Elizaveta Rivkina[10],
Bart E. van Dongen[2], and Helen M. Talbot[1]

[1]School of Civil Engineering and Geosciences, Newcastle University, UK
[2]School of Earth, Atmospheric and Environmental Sciences and Williamson Research Centre for Molecular Environmental Science, University of Manchester, UK
[3]Balıkesir University, Geological Engineering Department, Balıkesir, Turkey
[4]Earth, Ocean and Atmospheric Science, Florida State University, Tallahassee, FL, USA
[5]Department of Environmental Science and Analytical Chemistry (ACES) and the Bolin Centre for Climate Research, Stockholm University, Sweden
[6]Pacific Oceanological Institute Far Eastern Branch of the Russian Academy of Sciences, Russia
[7]International Arctic Research Center, University of Alaska, USA
[8]National Tomsk Research Polytechnic University, Russia
[9]GFZ German Research Centre for Geosciences, Helmholtz Centre Potsdam, Section 5.3 Geomicrobiology, Telegrafenberg, Potsdam, Germany
[10]Institute of Physicochemical and Biological Problems in Soil Science, Russian Academy of Sciences, Pushchino, Russia
[*]Now at: The Sir Charles Lyell Centre, Heriot-Watt University, Edinburgh, UK
[**]Now at: School of Science and the Environment, Manchester Metropolitan University, Manchester, UK
*Correspondence to:* J. Bischoff (j.bischoff@hw.ac.uk) and B. E. van Dongen (bart.vandongen@manchester.ac.uk)

**Abstract.** The Siberian Arctic contains a globally significant pool of organic carbon (OC) vulnerable to enhanced warming and subsequent release by both fluvial and coastal erosion processes. However, the rate of release, its behaviour in the Arctic Ocean and vulnerability to remineralisation is poorly understood. Here we combine new measurements of microbial biohopanoids including adenosylhopane, a lipid associated with soil microbial communities, with published glycerol dialkyl glycerol tetraether

5   (GDGTs) and bulk $\delta^{13}C$ measurements to improve knowledge of the fate of OC transported to the East Siberian Arctic Shelf (ESAS). The microbial hopanoid-based soil OC proxy $R'_{soil}$ ranges from 0.0 to 0.8 across the ESAS, with highest values near shore and decreases offshore. Across the shelf $R'_{soil}$ displays a negative linear correlation with bulk $\delta^{13}C$ measurements ($r^2$ = -0.73, p = <0.001). When compared to the GDGT based OC proxy, the Branched and Isoprenoid tetraether (BIT) index, a decoupled (non-linear) behaviour on the shelf was observed, particularly in the Buor-Khaya Bay where the $R'_{soil}$ shows limited

10   variation, whereas the BIT index shows a rapid decline moving away from the Lena River outflow channels. This reflects a balance between delivery and removal of OC from different sources. The good correlation between the hopanoid and bulk terrestrial signal suggests a broad range of hopanoid sources, both fluvial and via coastal erosion whilst GDGTs appear to be primarily sourced via fluvial transport. Analysis of ice complex deposits (ICDs) revealed an average $R'_{soil}$ of 0.5 for the Lena delta, equivalent to that of the Buor-Khaya Bay sediments, whilst ICDs from further East showed higher values (0.6-0.85). Al-

15   though $R'_{soil}$ correlates more closely with bulk OC than the BIT, our understanding of the endmembers of this system is clearly



still incomplete with east-west variations potentially reflecting differences in environmental conditions (e.g. temperature, pH) but other physiological controls on microbial BHP production under psychrophilic conditions are as yet unknown.

# 1 Introduction

The Arctic permafrost region is a globally significant organic carbon (OC) store containing approximately 1300 Pg (uncertainty range ∼1100 to 1500 Pg) of carbon. Approximately 800 Pg (60%) is stored below the ground in frozen permafrost with the remainder (∼500 Pg) occurring in non-permafrost soils, seasonally thawed in the active layer or in deeper taliks (Hugelius et al., 2014). Permafrost is defined as ground (soil or rock and includes ice and organic material) that remains below 0 °C for at least two consecutive years (van Everdingen, 2005) and is naturally particularly sensitive to an increase in global temperatures. It is therefore a focal point of on-going climate change research on the observed (e.g. Romanovsky et al., 2010) and predicted rise in atmospheric and soil temperature (IPCC, 2013). Rising temperatures in the Arctic cause, amongst other severe consequences for society and infrastructure, shifts in hydrological processes and progressive deepening and duration of permafrost thawing during the Arctic summer (Vonk et al., 2015). This destabilisation of permafrost deposits will increase the re-distribution of terrestrial organic matter (OM) to the Arctic Shelf and ultimately the Arctic Ocean by (1) transportation via the major Arctic rivers and (2) erosion of coastal areas.

The Arctic Ocean receives around 10% of the global river discharge while representing only 1% of the global ocean water body (Opsahl et al., 1999). Climate change has already increased the water discharge to the Arctic Ocean (Peterson et al., 2002). Arctic Rivers are distinct in their hydrologic regime with pronounced seasonality (Holmes et al., 2012, 2013). They discharge the majority of their annual load of water, sediment and total OC from May to July (Dittmar and Kattner, 2003; Holmes et al., 2012, 2013). The drainage basins of these rivers include areas of continuous and discontinuous permafrost (Feng et al., 2015; Gustafsson et al., 2011; Kotlyakov and Khromova, 2002, and references therein). Thawing of permafrost deposits is linked to a destabilization of stored carbon by top-down thawing at the active-layer/permafrost interface leading to collapse of ice-rich permafrost, also known as thermokarst, resulting in hydrological changes (Vonk et al., 2015). Thermokarst processes including massive erosional events can lead to increased mobility of old carbon (both dissolved and particulate) from the lower layers and strongly affects the balance of carbon dioxide ($CO_2$) and methane ($CH_4$) emissions from these environments (e.g. Gustafsson et al., 2011; Schuur et al., 2009; Vonk et al., 2015). For example, between 1985 and 2004 an increase in the proportion of mobilized terrestrial OC accounted for by ancient carbon of 3–6% has been estimated (Feng et al., 2013). This increased transport of older material will not only increase the release of $CO_2$ to the atmosphere (Drake et al., 2015; Mann et al., 2015; Spencer et al., 2015) but already observed increases in river discharge will also lead to increased terrestrial OC input into the Arctic Ocean (Savelieva et al., 2000; Semiletov et al., 2000, 2013). However, the fate of this terrestrial OC in the Arctic Ocean system is not well understood.

An additional source of OC to the Arctic Ocean is that stored frozen within coastal ice complex deposits (ICD). These deposits erode at a rate greater than that of temperate coasts with an average rate for the Arctic coast of $0.5 \, \mathrm{m \, y^{-1}}$ albeit with high local variability, up to $10 \, \mathrm{m \, y^{-1}}$ (Lantuit et al., 2013). The highest rates are found in the Laptev, East Siberian and



Beaufort Seas where the majority of the coast comprises frozen unlithified material highly susceptible to erosion (Figure 1; Lantuit et al., 2012, 2013). Although at present much of the unlithified coast is located in areas still largely protected by sea ice, continuing decline in sea ice extent will expose this material to erosion and increase sediment flux to the ocean (Lantuit et al., 2013). The relative contribution of permafrost ICD erosion to sedimentary carbon in the East Siberian Arctic Sea (ESAS)

is estimated to be $57 \pm 1.6\%$ (Vonk et al., 2012). Other recent estimates vary widely but suggest that between 15 to 66% of this carbon is remineralised to $CO_2$ producing a positive feedback to climate warming whilst the reminder is buried in shelf sediments (Knoblauch et al., 2013; Vonk et al., 2012). Furthermore, Tesi et al. (2016) showed that the potential for burial and degradation of terrestrial OC on the ESAS, based on the different chemical reactivity of different components, is dependent on the source material (Top-Soil vs. ICD) and the transportation pathways (river run-off vs. coastal erosion), highlighting the need

to better understand these sources.

A number of methods are used to trace different primary sources of sedimentary OM on the Arctic shelves including bulk $\delta^{13}$C, $\delta^{15}$N, C/N and molecular ratios (Cooke et al., 2009; Drenzek et al., 2007; Feng et al., 2013, 2015; Goñi et al., 2005; Guo et al., 2004; Gustafsson et al., 2011; Semiletov, 1999a, b; Tesi et al., 2014; van Dongen et al., 2008; Vonk et al., 2012). The latter include the BIT index (Hopmans et al., 2004), based on comparison of branched glycerol dialkyl glycerol tetraethers

(brGDGTs) from terrestrial soil environments and the isoprenoid GDGT crenarchaeol (e.g. De Jonge et al., 2014, 2015; Doğrul Selver et al., 2015; Sparkes et al., 2015), and the bacteriohopanepolyol (BHP) based $R'_{soil}$ index (De Jonge et al., 2016; Doğrul Selver et al., 2012, 2015).

BHPs are microbial membrane lipids comprising pentacyclic triterpenoids with an extended polyfunctionalised side chain (Rohmer, 1993, see Table S1 for structures). They occur in varying concentrations and compositions in a range of environmental

settings such as Arctic permafrost soils (Höfle et al., 2015; Rethemeyer et al., 2010), lakes (Coolen et al., 2008; Talbot and Farrimond, 2007; Talbot et al., 2003c), and marine sediments (e.g. Blumenberg et al., 2009, 2010, 2013; Cooke et al., 2009; Doğrul Selver et al., 2012; Talbot et al., 2014; Wagner et al., 2014; Zhu et al., 2011). Recent studies have indicated the potential of a specific group of BHPs with a cyclised side chain to be used as a tracer for soil organic matter (SOM) input in aquatic settings. Adenosylhopane (**1a**, see Table S1 for structures; Bradley et al., 2010, and references therein), two related structures

with yet undetermined terminal groups termed 'adenosylhopane type 2' (**1b**; Cooke et al., 2008a) and 'adenosylhopane type 3' (**1b'**; Rethemeyer et al., 2010) together with their C-2 methylated homologues (**2a**, **2b** and **2b'**, respectively) are common compounds in soils (Cooke et al., 2008a; Kim et al., 2011; Rethemeyer et al., 2010; Spencer-Jones et al., 2015; Xu et al., 2009; Zhu et al., 2011). However, these compounds are rarely found in marine settings with the exception of deep-sea fan systems, which comprise a significant proportion of terrestrial OM including BHPs (Cooke et al., 2008b; Handley et al., 2010; Wagner

et al., 2014). A BHP based SOM proxy was proposed in which the relative proportion of "soil marker" BHPs (adenosylhopane and related compounds) to the combined total of soil markers plus the commonly occurring BHP bacteriohopane-32,33,34,35-tetrol (BHT, **1f**) is calculated (Zhu et al., 2011). The use of BHT in this context is complicated as it is also found in varying proportions in soils but is frequently the most significant and, in some cases, only BHP in marine sediments hence its proposal as the only possible representative compound for marine OM dominated sediments (De Jonge et al., 2016; Zhu et al., 2011).



This proxy index, termed $R_{soil}$ (Eq. 1), constrains the proportion of terrestrial, soil derived material in marine sediments in a similar way to the BIT index (Hopmans et al., 2004).

The $R_{soil}$ index has been investigated in various setting including the Yangtze River-East China Sea surface sediment transect (Zhu et al., 2011) and several (sub-)Arctic land to ocean transects (De Jonge et al., 2016; Doğrul Selver et al., 2012, 2015) and showed that it can be used to trace SOM exported from land to ocean. However, due to the limited and intermittent occurrence of methylated compounds in the sub-Arctic setting, $R_{soil}$ (Eq. 1) was modified to $R'_{soil}$ for application in Arctic settings where the C-2 methylated soil markers are scarce and therefore were excluded (Eq. 2; Doğrul Selver et al., 2012, 2015).

$$R_{soil} = \frac{\text{soil BHPs}(\mathbf{1a + 1b + 1b' + 2a + 2b + 2b'})}{\text{soil BHPs}(\mathbf{1a + 1b + 1b' + 2a + 2b + 2b'}) + \text{BHT}(\mathbf{1f})} \tag{1}$$

$$R'_{soil} = \frac{\text{soil BHPs}(\mathbf{1a + 1b + 1b'})}{\text{soil BHPs}(\mathbf{1a + 1b + 1b'}) + \text{BHT}(\mathbf{1f})} \tag{2}$$

This study focuses on the East Siberian Arctic Shelf (ESAS), a region dominated by fluvial input from three major rivers; namely the Lena, Indigirka and Kolyma (Gordeev, 2006), as well as a site of significant erosion of coastal ICD (Figure 1; Lantuit et al., 2012). Recently, Sparkes et al. (2015) used the GDGT based BIT proxy to trace terrestrial OM on the ESAS shelf and found a decoupling between BIT and bulk $\delta^{13}$C, suggesting that GDGTs were (primarily) sourced via riverine transport and not from erosion of coastal ICD. This decoupling was also observed by Doğrul Selver et al. (2015) in sediments along the offshore transect off the Kolyma River, suggesting different and/or additional sources of BHPs to the ESAS compared to the GDGTs. Erosion of ICD was proposed as a likely source for the BHPs suggesting that these biomarker proxies probably reflect different OC sources (Doğrul Selver et al., 2015). It has, however, been suggested that coastal cliffs could also be a source of branched GDGTs, at least in sites without major river inputs (De Jonge et al., 2015). At this time, it remains unclear (i) if the decoupling between these bacterial biomarker based proxies is unique to the Kolyma region or more widely applicable to the whole ESAS and (ii) if the soil marker BHP pool has a mixed input from ICD and river transported OC or can be used as a proxy for ICD.

Therefore, this study investigates the abundance and composition of terrestrial microbial (soil marker) BHPs across the land – ocean transect of the ESAS in conjunction with the recently published BIT data (Sparkes et al., 2015). This study includes new data on BHPs in ICD from the Lena delta and Indigirka and Kolyma riverbanks in order to further constrain the source of OC transported to and deposited into the Arctic Ocean.

## 2 Materials and Methods

### 2.1 Study area and sample collection

This study focuses on the ESAS with sediment samples from the Laptev Sea, Buor-Khaya Bay, Dmitry Laptev Strait and East Siberian Sea. The comprehensive fieldwork was conducted in August – September 2008 as part of the International Siberian Shelf Studies 2008 expedition (ISSS 08; Semiletov and Gustafsson, 2009). Surface sediments were recovered using a dual



gravity corer or a van-Veen grab sampler from H/V Yakob Smirnitsky (ESAS) and TB-0012 (Buor-Khaya Bay). The sediment samples were transferred with stainless steel spatulas to polyethylene containers and frozen at -18 °C for transport and storage (Karlsson et al., 2011). Subsamples were taken and freeze-dried for subsequent total lipid extraction (Sparkes et al., 2015).

The sediments investigated in this study were grouped based on their location on the ESAS (Figure 1; Table S2; Sparkes et al., 2015). Samples were grouped longitudinally, into the Buor-Khaya Bay and associated region offshore the Lena River delta (the Laptev Sea), the Dmitry-Laptev Strait (the narrow channel between the coastline at ∼140°E and the New Siberian Islands, splitting the ESAS up into two distinct areas - the Laptev Sea and the East Siberian Sea), the region offshore of the Indigirka River mouth and the region offshore of the Kolyma River mouth. The Indigirka and Kolyma river mouth offshore regions are generally equivalent to the Western and Eastern East Siberian Sea regions, respectively, as identified by Semiletov et al. (2005). The ESAS samples have also been classified latitudinally, into the nearshore ESAS (<150 km from river outflows) and offshore ESAS (>150 km from river outflows).

In addition to surface sediment samples throughout the ESAS, this study also includes ICD samples from locations on the Siberian mainland, including central Lena Delta, Cape Bykovsky, the Kolyma and Indigirka river banks.

The site on Kurungnakh Island (central Lena Delta; 72°20'N, 126°17'E) was drilled during the Russian-German LENA 2002 expedition in July 2002 (Bischoff et al., 2013) and a 24 m long permafrost core from a low-centred ice-wedge polygon was recovered (Grigoriev, 2003). In total 23 samples from depths 0.34 to 24.55 m (Table S3) were chosen for BHP analysis. An additional ice complex sample (CB IC 1.9; Table S3) was obtained from from Cape Bykovsky. The Bykovsky Peninsula is located in the vicinity of the Lena Delta in an area of significant coastal erosion (Lantuit et al., 2011). Ice complex samples from the Kolyma region were obtained from the Chukochya River (CR; Figure 1), which outflows in the Kolyma Gulf, and The Omolon River (OR; Figure 1), a tributary of the Kolyma River, and from Cherskii (CH, Figure 1; Table S3; Tesi et al., 2014). An additional profile from the Indigirka watershed is also included for comparison (KY, Figure 1; Tesi et al., 2014).

## 2.2 Bulk analysis

Data for the ISSS-08 sediments is taken from Karlsson et al. (2011, see Table S2). TOC data for the Indigirka and Cherskii profiles are taken from Tesi et al. (2014, see Table S3). The Cape Bykovsky, Chukochya River and Omolon River permafrost ICD samples were prepared for carbon isotope analysis according to Harris et al. (2001) and analysed at the University of California, Davis, Stable Isotope Facility.

## 2.3 Extraction of ESAS sediment samples

Sediment samples were extracted using a modified Bligh-Dyer method as described in more detail in Doğrul Selver et al. (2015) and Sparkes et al. (2015). Briefly, sediment (5 g) was ultrasonically extracted using a monophasic mixture of methanol/ dichloromethane/phosphate buffer (0.05M, pH 7.4; 2:1:0.8 v/v/v). The supernatant was separated by centrifugation and the remaining sediments re-extracted twice. The combined organic phases were evaporated to dryness. After extraction, the total lipid extracts (TLE) were re-dissolved in dichloromethane/methanol (2:1) and separated into fractions with 1/3 of the TLE used for BHP analysis.





## 2.4 Extraction of ice complex samples

Freeze dried and ground ICD samples were extracted using a modified Bligh and Dyer method (1959) that was adapted from the method described in Cooke et al. (2008a). Briefly, samples (∼3 g, dry weight) were ultrasonically extracted with a monophasic mixture of methanol/chloroform/water (10:5:4 v/v/v). Deviating from the method described in Cooke et al.

(2008a), the sonication steps were reduced to 30 min at 40°C without the subsequent overnight shaking. The supernatant was removed after centrifugation (12,000 rpm, 10 min) and the remaining sediment was re-extracted twice. After phase separation via addition of water (5 mL) and chloroform (5 mL), the organic phases were combined, evaporated to dryness, and blown to dryness under $N_2$. Extracts were then re-dissolved in 2:1 chloroform/methanol and 1/3 of the TLE was used for BHP analyses.

## 2.5 Solid phase extraction and derivatisation of BHPs

Aliquots (1/6 for sediments and 1/3 for terrestrial materials) of the TLE were loaded onto $NH_2$ solid phase extraction (SPE) cartridges pre-conditioned with 6 mL hexane (1 g/6 mL; Isolute, Biotage, Sweden), in 200 µL chloroform and separated into 2 fractions (Fr): Fr. 1 (non-polar + acidic, 6 mL diethylether/acetic acid [98:2, v:v]) and Fr. 2 (polar, 12 mL methanol) which contained all BHPs except 32,35-anhydroBHT (e.g. Bednarczyk et al., 2005). After separation, the internal standard (5α-pregnane-3β,20β-diol) was added to Fr. 2 and dried under nitrogen. This SPE method was adapted from a method commonly

used in other studies of complex polar lipids from environmental samples (e.g. Lupascu et al., 2014).

Fr. 2 was acetylated with pyridine/acetic anhydride (1:1, v:v; 500 µL) for 1 h at 50 °C and left at room temperature overnight. The samples were evaporated to dryness, re-dissolved in methanol/propan-2-ol (60:40, v:v) and filtered through a 0.2 µm PTFE syringe filter. For BHP analysis, the samples were dissolved in methanol/propan-2-ol (60:40, v:v; 500 µL). Sample injection volume was 10 µL.

## 2.6 Analytical HPLC-APCI-MS

BHPs were identified and measured using reverse phase HPLC-APCI-MS as previously described in Cooke et al. (2008a). Chromatographic separation was achieved under the conditions described in van Winden et al. (2012b). BHP structures were identified based on previously published spectra (Cooke et al., 2008a; Rethemeyer et al., 2010; Talbot and Farrimond, 2007; Talbot et al., 2003a, b).

Semi-quantitative estimation of BHP concentration was achieved by employing the characteristic base peak ion areas of individual BHPs in mass chromatograms (from SCAN 1) relative to the m/z 345 chromatogram base peak area of the acetylated 5α-pregnane-3β,20β-diol internal standard. Averaged relative response factors relative to the internal standard, determined from a suite of acetylated BHP standards, were used to adjust the BHP peak areas where N containing compounds give an average response 12 times than of the standard and compounds without N 8 times that of the standard (for further details see

van Winden et al., 2012b). The reproducibility of triplicate injections was 3-6% RSD (standard error: $\pm 1 - 4\ \mu g\ g_{OC}^{-1}$) for BHT and 5-8% RSD (standard error: $\pm 1 - 2\ \mu g\ g_{OC}^{-1}$) for adenosylhopane in the environmental samples, resulting in an absolute standard error of on average ±0.01 for $R'_{soil}$ (see Section 1, Equation 2).





## 3   Results and Discussion

OC concentrations and bulk carbon isotopes for sediments recovered throughout the ISSS-08 expedition have been reported previously (Vonk et al., 2012). OC concentrations ranged from 0.68 to 2.25 wt. %C and were highest in Buor-Khaya Bay with values for the different regions are shown in Table S1 (see also Sparkes et al., 2015).

### 3.1   BHP Concentrations and distributions in ESAS surface sediments

In total, 92 surface sediment samples throughout the ESAS in Bour-Khaya Bay, Dimitry Laptev Strait and Kolyma and Indigirka river mouth transects were analysed for BHP composition (Table S2).

Up to 16 individual BHPs were identified in the ESAS sediments, with the total concentration of BHPs ranging from 12 – 824 $\mu g\,g_{OC}^{-1}$ (Table S2). However, their concentrations and distributions differed with distance to the mainland throughout the shelf. BHT (**1f**) was the most abundant single BHP ranging from 9 – 313 $\mu g\,g_{OC}^{-1}$ (Table S1, Figures 2a and 3a). The relative proportion of BHT was lowest in Buor-Khaya Bay sediments close to the mainland (mean = 37 % of all detected BHPs) rising to 80 % (mean = 65 %) of all detected BHPs in the ESAS offshore sediments furthest away from the mainland. Highest BHT concentrations were measured closest to the mainland except in the region of the Indigirka outflow (samples YS-26 to 30; Table S2) where values were lower than all other nearshore settings (Figure 2a). However, mean concentrations were stable with increasing distance from the mainland (Figure 3a) and considerable amounts of BHT (up to 77 $\mu g\,g_{OC}^{-1}$) are still detectable at 293 km offshore.

In addition to the ubiquitous and abundant BHT (**1f**), a suite of other polyhydroxylated BHPs related to BHT were detected, including BHT-isomer (**1f'**) which has been linked to production by pelagic anaerobic organisms performing anaerobic ammonium oxidation (annamox; Rush et al., 2014). The C-2 methylated homologue 2-MeBHT (**2f**), unsaturated BHT ($\Delta^6$-**1f**) and bacteriohopane-30,31,32,33,34,35-hexol (**1g**) were also common, especially in Buor-Khaya Bay, although at much lower concentration than BHT (Table S2).

Soil marker BHPs identified included high proportions of adenosylhopane (**1a**), followed by adenosylhopane type 2 (**1b**) and adenosylhopane type 3 (**1b'**). The soil marker type 2 and 3 compounds are related to adenosylhopane but have different and as yet uncharacterised terminal groups compared to adenosylhopane as identified by LC-MS$^n$ (Table S1; Rethemeyer et al., 2010). The C-2 methylated soil markers (**2a**, **2b**, **2b'**; Table S1) were present intermittently and always at lower concentration than the corresponding non-methylated structures. Generally, the concentration of non-methylated soil markers are highest in samples closer to the coast (0-100 $\mathrm{km}$) and decrease with distance from the river outflows (Figures 2b and 3b; Table S2).

Sediments of Buor-Khaya Bay and Dmitry-Laptev strait were characterised by high amounts of adenosylhopane, with a mean average of 64 $\mu g\,g_{OC}^{-1}$ (range 7 – 137 $\mu g\,g_{OC}^{-1}$; Table S2) and total non-methylated soil markers accounted for up to 66% of total BHPs although the mean proportion was lower at 36%. Sediments collected from the ESAS offshore region contain noticeably lower soil marker BHPs both in absolute and relative concentrations with non-methylated soil markers ranging from 0 – 62 $\mu g\,g_{OC}^{-1}$ (mean 11 % of all detected BHPs; Table S2; Figures 2b and 3b). Methylated compounds were frequently



below the detection limit (Table S2) as previously reported in Arctic and sub-Arctic River mouth surface sediment transects (Doğrul Selver et al., 2012, 2015).

Total concentrations of non-methylated soil markers in ESAS sediments with a range of 0 - 218 $\mu g\, g_{OC}^{-1}$ are similar to sediments from the Yenisei River system including the Yenisei River mouth, gulf, outflow and nearby Kara Sea (9 – 508 $\mu g\, g_{OC}^{-1}$; De Jonge et al., 2016). The highest values occurred in the Yenisei River mouth sediments and also in Khalmyer Bay (329 - 435 $\mu g\, g_{OC}^{-1}$), a nearby area of coastal erosion, although not directly drained by the Yenisei River itself. However, in both the Yenisei mouth, Khalmyer Bay, Buor-Khaya Bay and Dmitry-Laptev Strait, non-methylated soil markers had very similar mean relative abundance as a proportion of total BHPs of 30-40%. This reflects a significant contribution of these terrestrial compounds to the total BHP assemblage in Arctic settings (see also Cooke et al., 2009; Taylor and Harvey, 2011) and indicates a strong terrestrial signal in the Arctic Shelf sediments. The proportion of total soil markers in sediments from the outflow of other non-Arctic rivers has been shown to be rather lower (e.g. <20% from the estuary and seaward of the Yangtze River, Zhu et al., 2011; <8% Congo River estuary, Spencer-Jones et al., 2015) thus emphasising the importance of obtaining values for local endmembers. The higher abundance of these highly functionalised compounds in Arctic sediments may be a result of better preservation under the cold temperature conditions of this region. Additionally, temperature may also influence the microbiological community leading to deliberate accumulation of adenosylhopane and/or limited biosynthetic transformation of the BHP precursor in this extreme environment (cf. Rethemeyer et al., 2010).

Other BHPs identified include three structures with amine functional groups at the C-35 position. The concentration of aminotriol (**1e**; Table S1) varied from 0 - 53 $\mu g\, g_{OC}^{-1}$ (mean 6.4 % of all analysed BHPs) throughout the ESAS (Table S2). Aminotetrol (**1d**) was generally less abundant across the ESAS (0 – 13 $\mu g\, g_{OC}^{-1}$) and Aminopentol (**1c**) was identified in 37 of the 47 Buor-Khaya Bay sediments and only occasionally in other areas (Table S2). Aminotetrol and aminopentol in particular have been linked to aerobic methane oxidising bacterial sources and have been proposed as markers for terrestrial methane oxidation in continental wetlands, which is then subsequently recorded in deep-sea fan sediments from the Congo (Spencer-Jones et al., 2015; Talbot et al., 2014; Wagner et al., 2014). More recently, De Jonge et al. (2016) identified aminopentol in sediments of the Yenisei River outflow and tentatively proposed that it might indicate decomposition of sub-sea permafrost with associated methane release and subsequent microbial oxidation. Here in Buor-Khaya Bay, it is possible that the aminopentol signature is either fluvially transported from areas of active aerobic methane oxidation within the catchment (i.e. wetlands, lakes) or alternatively indicates oxidation of methane released from sub-sea permafrost deposits known to be common across the ESAS including Buor-Khaya Bay (Shakhova and Semiletov, 2007; Shakhova et al., 2009, 2010a, b, 2014, 2015).

The ESAS surface sediments generally had only low levels of composite BHPs (i.e. BHPs with more complex moiety at the C-35 position such as a sugar or amino-sugar; e.g. Rohmer, 1993). The only exceptions were BHT cyclitol ether (**1h**) which was found in concentrations ranging from 0 - 49 $\mu g\, g_{OC}^{-1}$, mean 7.7 $\mu g\, g_{OC}^{-1}$ (0 - 10% of all analysed BHPs), and BHT glucosamine (**1i**) which was even less common (Table S2). Both of these structures have a wide range of known sources so cannot be assigned to any specific group of source organisms (see review in Talbot et al., 2008).





## 3.2 $R'_{soil}$, stable carbon isotopes and BIT in ESAS sediments

Bulk carbon isotope values ($\delta^{13}$C) are commonly used as a proxy for marine vs. terrestrial influence on sedimentary OC composition as terrestrial plants using the C3 synthesis pathway typically have more depleted values than OC produced via marine primary productivity (Hayes, 1993; Meyers, 1997; van Dongen et al., 2008). Here, we compared bulk carbon isotopes

to the BHP based $R'_{soil}$ proxy (Equation 2). As expected, $R'_{soil}$ values were higher closer to the coast and reduced gradually with increasing distance offshore (Figures 2c, 3c) including the region off the Indigirka which had somewhat lower absolute concentrations of the individual BHPs compared to the more Eastern and western extents of the ESAS region (Figures 2a, 2b). The highest values occurred in Buor-Khaya Bay (maximum $R'_{soil}$ = 0.80; Table S2), however, the mean values for Buor-Khaya Bay and Dmitry-Laptev Strait sediments were very similar at 0.49 and 0.52 respectively (Figure 4). The ESAS nearshore

sediments had an average only slightly lower at 0.41 whilst the average for the offshore sediments was 0.14 (range 0.42 to 0.00; Figure 4, Table S2).

A clear negative linear relationship was observed between $R'_{soil}$ and bulk $\delta^{13}$C ($r^2$ = -0.73, p <0.001; Figure 5a) across the ESAS in agreement with a pilot study of soil-microbial biomarkers in ESAS surface sediments from the Kolyma River mouth offshore transect where strong linear correlations were observed between the $R'_{soil}$ proxy and distance from river mouth ($r^2$

= 0.97) and bulk $\delta^{13}$C ($r^2$ = 0.96; Doğrul Selver et al., 2015). Although bulk $\delta^{13}$C in ESAS surface sediments did display a linear relationship with distance offshore (Vonk et al., 2012), previous comparison of bulk $\delta^{13}$C and BIT values from the ESAS surface sediments displayed a strongly non-linear correlation (Figure 5c; Sparkes et al., 2015). This was shown to result from a rapid reduction in concentration of brGDGTs in near coastal sediments (<150 km from shoreline) causing a drop in BIT values from 1 to 0.25 with little variation in bulk $\delta^{13}$C followed by an enrichment in bulk carbon isotopes towards more

marine values concomitant with a slower decline in BIT values towards 0 further than 150 km offshore (Sparkes et al., 2015). A similar non-linear relationship is observed here between $R'_{soil}$ and BIT (Figure 5b), as also previously demonstrated for the surface sediment offshore transect off the Kolyma River mouth (Doğrul Selver et al., 2015), indicating that similar processes are operating across the entire ESAS.

The simple linear correlation between $R'_{soil}$ and bulk carbon isotope values is intriguing as it suggests that, unlike the

brGDGTs which in this region are proposed to be primarily derived from fluvial transport (De Jonge et al., 2014; Doğrul Selver et al., 2015; Peterse et al., 2014; Sparkes et al., 2015), the $R'_{soil}$ proxy provides a more integrated signature of different terrestrial sources including ICD and fluvially transported topsoil-permafrost or riverine produced material. Therefore, soil marker BHPs and brGDGTs, despite being nominally derived from similar sources i.e. terrestrial microbial membrane lipids, appear in fact to be representing different aspects of terrestrial OC export.

De Jonge et al. (2016) recently demonstrated that soil marker BHPs can indeed be transported in suspended particulate matter (SPM) from the Yenisei River, a large river located west of the ESAS. In the Yenisei study, only a moderate correlation was observed between the $\delta^{13}$C values and $R'_{soil}$ ($r^2$ = 0.44, p<0.01) suggesting they trace different pools of OM (bulk terrigenous OM versus bacterial OM). However, unlike in the current study, De Jonge et al. (2016) also found a strong linear correlation between $R'_{soil}$ and BIT ($r^2$ = 0.82, p<0.05). The Yenisei River catchment and outflow have significantly different characteristics





to those of the rivers entering the ESAS, specifically in terms of the extent of different permafrost regimes. Permafrost is classified as isolated, sporadic, discontinuous and continuous according to its spatial distribution (e.g. Zhang et al., 2008). Continuous permafrost means that over 90% of the area is frozen in contrast to discontinuous permafrost where only 30 – 80 % of the area is underlain by permafrost. The Yenisei drains an area with a high proportion of discontinuous permafrost (55%)

whilst the proportion of continuous permafrost is lower (33%; Feng et al., 2013, and references therein). The proportion of continuous permafrost in the Eastern river catchments is higher, ranging from 79% (Lena) to 100% (Kolyma and Indigirka; Feng et al., 2015, and references therein). This may in turn affect the composition and preservation of terrestrial microbial markers. Indeed, previous work has indicated that OC from continuous permafrost areas is older but also less degraded and more biolabile than that from areas of discontinuous permafrost (e.g. Cooke et al., 2009; Feng et al., 2015; Mann et al.,

2015; Spencer et al., 2015; van Dongen et al., 2008). Additionally, although there is some evidence of coastal erosion of ICD in this region, such as in the Khalmyer Bay area, this is not directly drained by the Yenisei River. Therefore, in this more westerly region it is likely that the primary source for both soil marker BHPs and brGDGTs is fluvial transport, hence the linear relationship between soil microbial biomarkers and bulk $\delta^{13}$C values (De Jonge et al., 2015, 2016). Whereas in the more easterly region we assume brGDGTs to be of fluvial origin (Sparkes et al., 2015) and the soil marker BHPs to have an integrated

signature of terrestrial sources including ICD and fluvially transported topsoil-permafrost or riverine produced material.

### 3.3   Terrestrial endmembers

Given the apparent discrepancy between the BHP and GDGT-derived signals, further consideration of the terrestrial endmembers is clearly required. Although extensive databases exist for different bulk isotopic endmembers for the Arctic Region (e.g. Tesi et al., 2014; Vonk et al., 2012, and references therein), data on the soil microbial lipid derived proxy values are limited for

the Siberian region (see Table 1 for summary). Recently Peterse et al. (2014) reported high BIT values for a range of materials from the Kolyma region (Eastern ESAS) including thermokarst and floodplain lake sediments (BIT = 1), yedoma (and associated streams; BIT = 0.81 – 0.89) and SPM from the Kolyma River including samples collected during the spring freshet (BIT = 0.99 – 1). However, Sparkes et al. (2015) reported values for 3 ICD (Yedoma) samples from the same area ranging from 0.44 – 0.7. The lower values resulted from relatively high levels of crenarchaeol which is unusual for terrestrial materials (typical BIT

values >0.8; Schouten et al., 2013), although this compound has been reported from several Thaumarchaeota isolated from soil (Sinninghe Damsté et al., 2012). Data on BHPs from this region is scarce as previous studies of terrestrial BHPs have primarily focussed on temperate and more recently on tropical regions (see review in Spencer-Jones et al., 2015). Doğrul Selver et al. (2015) reported average $R'_{\text{soil}}$ values of 0.76 (range 0.70 – 0.84) for the same 3 ICD yedoma samples from the Kolyma region (CHYED-2; Table 1, Table S3). Höfle et al. (2015) also reported the BHP composition in polygonal active layer deposits (to a

maximum depth of 48 cm) from two locations in the Lena delta, Samoylov Island and Kurungnakh Island. Calculating $R'_{\text{soil}}$ values from this data revealed a wide range of values from 0.18 to 0.79 and a mean average of 0.41 (Table 1). Although this simple average will likely not represent a spatially and depth resolved average for the region, it is still close to the mean values found in Buor-Khaya Bay and Dmitry-Laptev Strait sediments (Table S2).



To further evaluate potential endmember ranges, additional ICD samples from the Lena, Indigirka and Kolyma regions were investigated for BHPs (Table S3). We analysed 23 samples from a 25 m permafrost core from Kurungnakh Island (see details in Bischoff et al., 2013) and an additional sample from Cape Bykovsky at 1.9 m depth (Table S3). BHT and a range of soil marker BHPs were present in all samples. BHT concentration ranged from 7 to 643 $\mu g\,g_{OC}^{-1}$ (mean avg. 248 $\mu g\,g_{OC}^{-1}$) in the

Kurungnakh Island permafrost deposits. As in the ESAS sediments (Table S2), the soil marker BHPs included high proportions of adenosylhopane (1a), followed by adenosylhopane type 2 (1b) and adenosylhopane type 3 (1b') (mean average 250 $\mu g\,g_{OC}^{-1}$; Table S3). As expected, observation of the methylated compounds was intermittent and then only at very low levels (Table S3) justifying their exclusion from $R'_{soil}$ (Doğrul Selver et al., 2012, 2015). $R'_{soil}$ values ranged from 0.37 to 0.64 with a mean value of 0.50 (Table 1) whilst the Cape Bykovsky (CB) sample had an $R'_{soil}$ of 0.68 (Table 1). The low $R'_{soil}$ values in ICD

from this region (0.34 to 0.80, mean 0.49; Figures 3c, 4) are in excellent agreement with the mean and range of values found within the ISSS-08 Buor-Khaya Bay and Dmitry-Laptev Strait sediments (0.49 and 0.52 respectively; Table S2; Figure 5a). Although additional sources from fluvial transport and from material transported via changes in hydrological conduits resulting from thermokarst erosion are also possible (e.g. Vonk et al., 2015), there is currently no data on the BHP composition of these materials from this region for comparison.

Bulk $\delta^{13}C$ was not measured for the KUR core used in this study but Wagner et al. (2007) reported values between -23.06 and -24.63 ‰for the OC fraction of selected permafrost sediments from Samoylov Island which lies close to Kurungnakh Island in the Lena Delta (see map in Höfle et al., 2015). These values from Samoylov Island are significantly enriched relative to the value for the Cape Bykovsky sample (-25.99 ‰; Table S3) and may reflect input from aquatic plants in low-centre polygon ponds (Schirrmeister et al., 2011). However, input from peat, grasses, herbs and shrubs result in more negative values such as

those reported by Schirrmeister et al. (2011) for a range of sites in the region including Kurunganakh Island and the Bykovsky peninsula (range -30 to -25 ‰). The $\delta^{13}C$ values for the ISSS-08 sediments from Buor-Khaya Bay (-25.3 to -26.6 ‰) and the Dmitry-Laptev Strait (-26.9 to -27.4 ‰) therefore suggest a significant contribution of terrestrial derived material. Vonk et al. (2012) reported an extensive compilation of circum-Arctic literature data with an average $\delta^{13}C$ value of -26.3 $\pm$ 0.67 ‰for ICD OC (coastal, inland and subsea; formed before inundation) and even more depleted values for topsoil permafrost with

$\delta^{13}C$ of -28.2 $\pm$ 1.96 ‰ (Table 1). By combining bulk $^{13}C$ and $^{14}C$ data, these authors estimated the proportion of sedimentary OM derived from ICD, topsoil and marine sources with over two-thirds of the OM in Buor-Khaya Bay derived from terrestrial sources and even higher values in Dmitry-Laptev Strait sediments.

ICD samples from the Indigirka (n = 3) and Kolyma River (n = 8) regions had lower absolute concentrations of BHPs (BHT range 8.5 to 62 $\mu g\,g_{OC}^{-1}$, non-methylated soil markers range 43 to 123 $\mu g\,g_{OC}^{-1}$) and generally higher $R'_{soil}$ values than those

from the Lena region (mean 0.76, range 0.62 – 0.84; Tables 1, S3; Figure 5a). Given the higher $R'_{soil}$ values for ICD in the eastern region, this suggests that although some removal of soil marker BHPs may have already occurred in the river estuaries before reaching the near coastal shelf sediments, a significant proportion remains intact. Based on the average ICD permafrost endmember data for the Kolyma and Indigirka samples this would suggest that ∼75% of the OC at the start of the offshore Kolyma River mouth surface sediment transect (sample YS-34B, $R'_{soil}$ = 0.57) and ∼63% at the start of the offshore transect

off the Indigirka River mouth (sample YS-30, $R'_{soil}$ = 0.48) is of terrestrial origin. These values should be treated with caution





given the limited number of samples involved, however, the value for the Indigirka in particular is close to the average value reported for the ESAS surface sediments based on dual-carbon-isotope ($\delta^{13}$C and $\Delta^{14}$C) mixing models (57±1.6% from ICD; Vonk et al., 2012).

This increase in $R'_{soil}$ values in permafrost from west to east is also noteworthy as movement from west to east represents a

change from discontinuous to fully continuous permafrost conditions within the catchment and to colder and drier conditions (Gordeev, 1996), potentially indicating better preservation of the structurally more complex soil marker BHPs relative to BHT (see section 3.2). This agrees with previous studies which have also found better preservation of more highly functionalised and biolabile molecules in materials from the eastern region (e.g. Guo et al., 2004; van Dongen et al., 2008). Tesi et al. (2016) reported that fine and ultrafine grain size fractions contain a high proportion of high molecular weight lipid markers which are

preferably bound to the mineral matrix and that the reactivity of lipid biomarkers on the ESAS seems to be lower and inversely proportional to the number of functional groups (cutin acids > n-alkanolic acid > n-alkanols > n-alkanes). Even though the reactivity for different BHPs is currently unknown, this points towards a potential recalcitrance of highly-functionalised BHP molecules on the ESAS. Furthermore, studies from soils have shown the potential for mineral-organic interactions, leading to increased resistance to degradation for aromatic compounds (Mikutta et al., 2007, 2009). Adenosylhopane, the most abundant

single soil marker BHP on the ESAS, is the only BHP containing an aromatic moiety (adenine) and therefore organic-mineral interactions may be a factor explaining the high relative abundance of these compounds under certain conditions.

However, the current sample set discussed is limited given the enormous spatial scale and extremely heterogeneous nature of these environments and does not include, for example, material released from thermokarst environments including thermokarst lake sediments which can be an important source of OC and inorganic material (Vonk et al., 2015). Furthermore, environmental

parameters other than location (and inferred temperature) must also be considered as potentially affecting the overall BHP assemblage. For example, Höfle et al. (2015) recently demonstrated using principal component analysis that increasing pH was positively correlated with soil marker BHP concentration in Lena delta permafrost. These authors proposed that this might indicate source organisms do not need to further extend their BHP side chains to alter membrane architecture at near neutral conditions. Furthermore, studies of temperate *Sphagnum* peat deposits, which typically have low pH values, show low

abundance of soil marker compounds relative to total BHPs when compared to mineral soils, but have higher abundance of BHT resulting in very low $R'_{soil}$ values (mean 0.4; calculated from data in van Winden et al., 2012a, b). This could suggest that in areas with a significant input from peat-derived material in some areas/layers of the Lena delta ICD (Bischoff et al., 2013; Wagner et al., 2007), lower $R'_{soil}$ values should be expected in agreement with observations in this study (Table S3). Indeed pH has also been shown to play a dominant role in shaping bacterial communities with the capacity to produce hopanoids in

an acidic peatland (Gong et al., 2015). Clearly a more comprehensive assessment of different terrestrial endmembers across the region is required as are additional studies on primary environmental factors affecting BHP biosynthesis in culture. For example to date no studies have investigated biosynthesis of adenosylhopane in psychrophilic/psychrotolerant organisms at different temperatures or at different growth stages, and studies of BHPs in association with pH adaptation have yet to measure adenosylhopane abundances (e.g. Welander et al., 2009). Given that adenosylhopane is the precursor for biosynthesis of all

other side chain extended BHPs (Bradley et al., 2010), it is possible that under conditions of stress (such as extreme temperature



and nutrient limitation), production of adenosylhopane without further modification is sufficient and/or all that some organisms are capable of and further modification is metabolically unfavourable/unnecessary.

## 4   Conclusions

Different suites of terrestrial microbial membrane lipids (biohopanoids and brGDGTs) and bulk carbon isotopes were used to trace the source and transport of terrestrial OC on the ESAS. As expected, ESAS sediments are terrestrially dominated; however, BHP and GDGT based SOM proxies are decoupled in Buor-Khaya Bay, SE Laptev Sea and across the ESAS, in agreement with an earlier pilot study of the surface sediment offshore transect off the Kolyma River mouth (Doğrul Selver et al., 2015). This is likely due to different sources, transport and/or degradation pathways for the various lipids. In particular, whilst brGDGTs have previously been shown to be primarily delivered to the ESAS via fluvial transport (Sparkes et al., 2015), BHPs appear to provide a more integrated signature correlating linearly with bulk carbon isotope ratios as well as distance from river mouths. The BHP terrestrial endmembers, Adenosylhopane and other soil marker BHPs, are significant components of coastal ICD which is transferred to ESAS sediments during coastal erosion. $R'_{soil}$ proxy values, although still limited for this region, vary widely with on average significantly lower values occurring to the Western range of East Siberia (average 0.5 for the Lena delta ICD) and higher values further east (average 0.76 for Indigirka and Kolyma ICD). The controlling factors responsible for this difference may include the transition from discontinuous to continuous permafrost (west to east), however, other factors such as mineral grain size, temperature (including annual minimum – maximum range), precipitation and pH as well as microbial/metabolic factors under psychrophilic conditions require further investigation.

*Author contributions.* Ö. Gustafsson, B. E. van Dongen, O. V. Dudarev, and I. P. Semiletov collected samples along with the crew of ISSS-08. Ice-complex samples were collected and provided by R.G.M. Spencer, E. Rivkina, D. Wagner and A. N. Kurchatova. H. M. Talbot and B. E. van Dongen designed the study, which was carried out by J. Bischoff, with assistance from R. B. Sparkes and A. Doğrul Selver. H. M. Talbot, J. Bischoff, R. B. Sparkes and B. E. van Dongen prepared the manuscript with contributions from all co-authors.

*Acknowledgements.* We gratefully acknowledge receipt of a NERC research Grant (NE/I024798/1 and NE/I027967/1) to B. E. van Dongen and H. M. Talbot., a Ph.D. studentship to A. Doğrul Selver funded by the Ministry of National Education of Turkey, financial support as an Academy Research Fellow to Ö.G. from the Swedish Royal Academy of Sciences through a grant from the Knut and Alice Wallenberg Foundation and support from the Government of the Russian Federation (Grant #14, Z50.31.0012/03.19.2014) to I.S. and from the Russian Scientific Foundation to O.D. (Grant # 15-17-20032). We thank the crew and personnel of the R/V Yakob Smirnitskyi and all colleagues in the International Siberian Shelf Study (ISSS) Program for support, including sampling. We thank A. N. Kurchatova for assistance with fieldwork on Kurungnakh Island and T. Tesi for providing the Yedoma samples for the Kolyma and Indigirka catchment areas We thank P. Lythgoe (University of Manchester) and F. Sidgwick (Newcastle University) for invaluable assistance with LCMS and the Science Research Investment Fund (SRIF) From HEFCE for the ThermoFinnigan LCQ ion trap mass spectrometer (Newcastle University). R.G.M. Spencer was partially supported by the U.S. National Science Foundation (ANT-1203885/PLR-1500169). The ISSS program, is supported by the



Knut and Alice Wallenberg Foundation, the Far Eastern Branch of the Russian Academy of Sciences, the Swedish Research Council, the US National Oceanic and Atmospheric Administration, the Russian Foundation of Basic Research, the Swedish Polar Research Secretariat, the Nordic Council of Ministers and the US National Science Foundation. Finally, we thank the associate editor and XXXXXX reviewers for constructive suggestions.




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




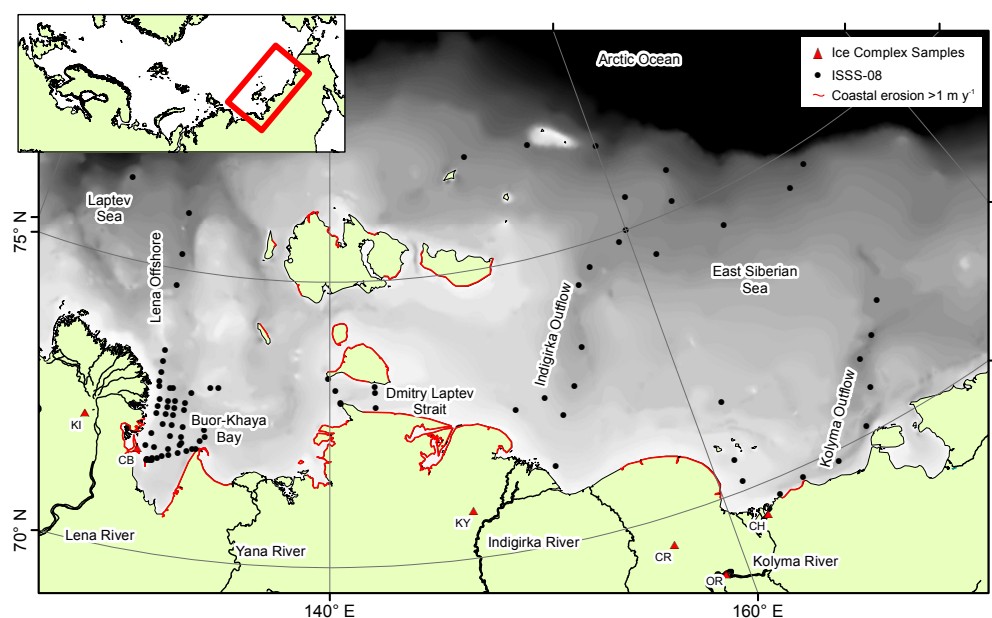

**Figure 1.** Map of the East Siberian Arctic Shelf (ESAS) showing sampling stations of International Siberian Shelf Studies 2008 (ISSS-08) expedition and location of Ice complex deposit (ICD) samples investigated in this study. Key regions discussed in the text are highlighted. Lower courses and outflows of four Great Russian Arctic Rivers are labelled. Section of coastline indicated in Red are areas of moderate to high rates of coastal erosion (>1 m y-1) as defined by (Lantuit et al., 2011). Key: KY = Kurungnakh Island; CB = Cape Bykovsky; KY = Indigirka, (Tesi et al., 2014); CR = Chukochya River; OR = Omolon River; CH = Cherskii (Tesi et al., 2014).





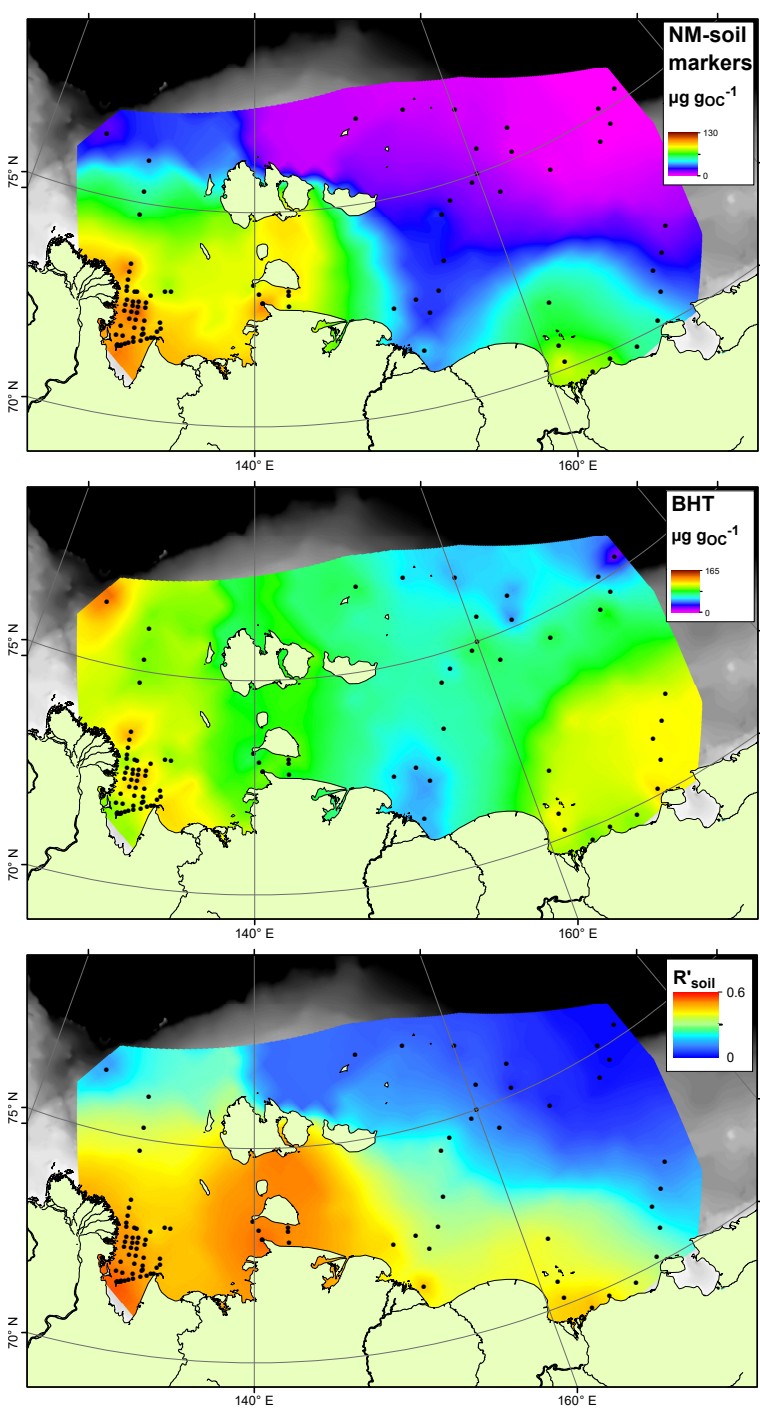

**Figure 2.** Maps of (a) BHT and (b) summed non-methyl soil BHP concentration (µg g$_{OC}^{-1}$) and (c) the resulting $R'_{soil}$ in ISSS sediments from ESAS. Maps were interpolated using a kriging algorithm (ArcGIS v.10; ESRI Ltd) and the locations of the ISSS-08 stations are shown as black dots.



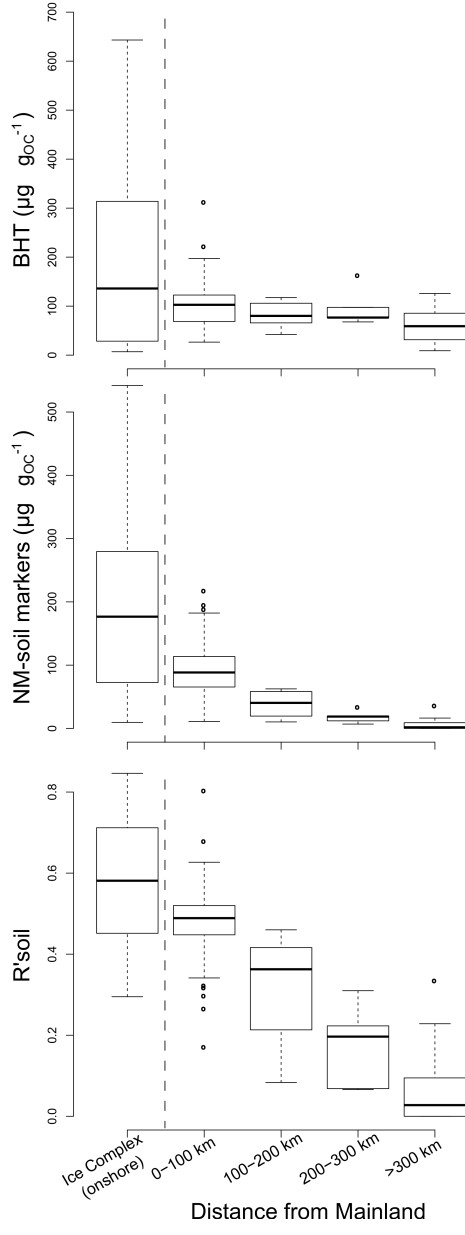

**Figure 3.** Box plots summarising the concentrations of (a) BHT , (b) summed non-methyl soil BHP concentration and (c) the resulting $R'_{soil}$ on the ESAS, grouped by distance from river mouths. Concentrations in ice complex samples are also shown (see Figure 1 for locations). Thick lines show the median values, boxes the 25th and 75th percentiles, whiskers the maximum and minimum values within 1.5 times the inter-quartile range and circular symbols outliers beyond this threshold.




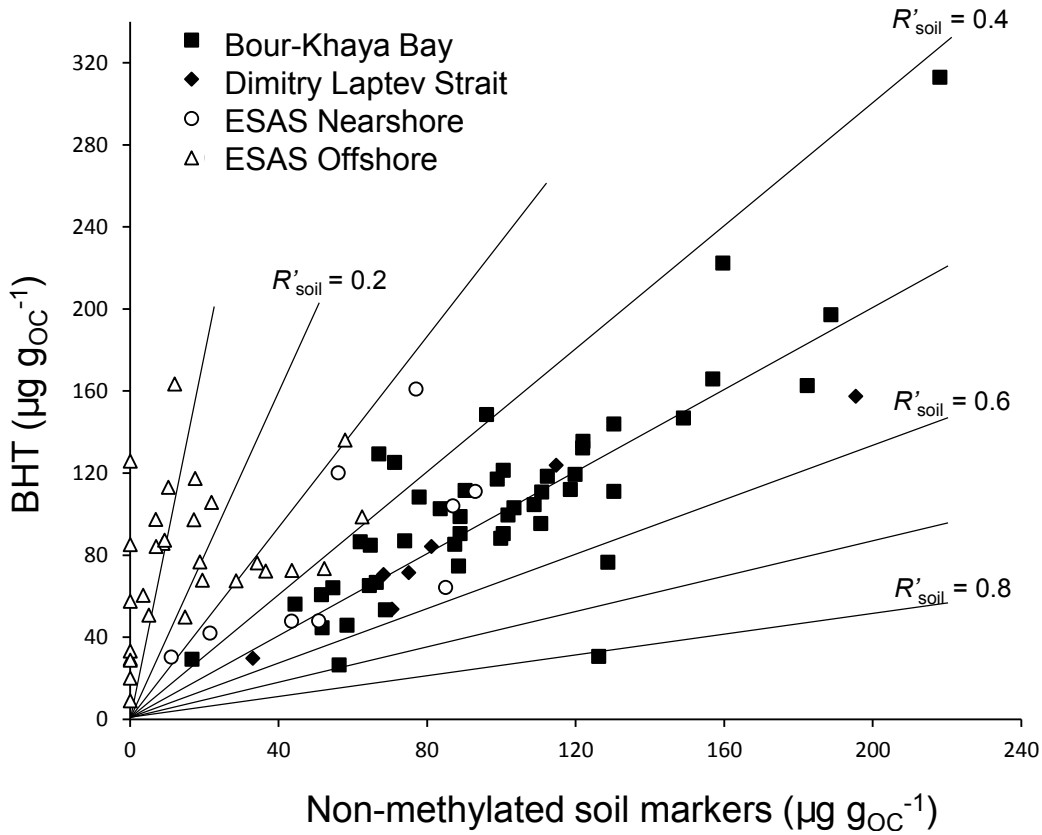

**Figure 4.** Plot of the concentration ($\mu g\ g_{OC}^{-1}$) of BHT vs. non-methyl soil BHPs grouped according to sampling location in Buor-Khaya Bay, Dmitri Laptev Strait, ESAS near shore and off shore. Labelled contours show the $R'_{soil}$ index values.

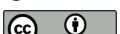



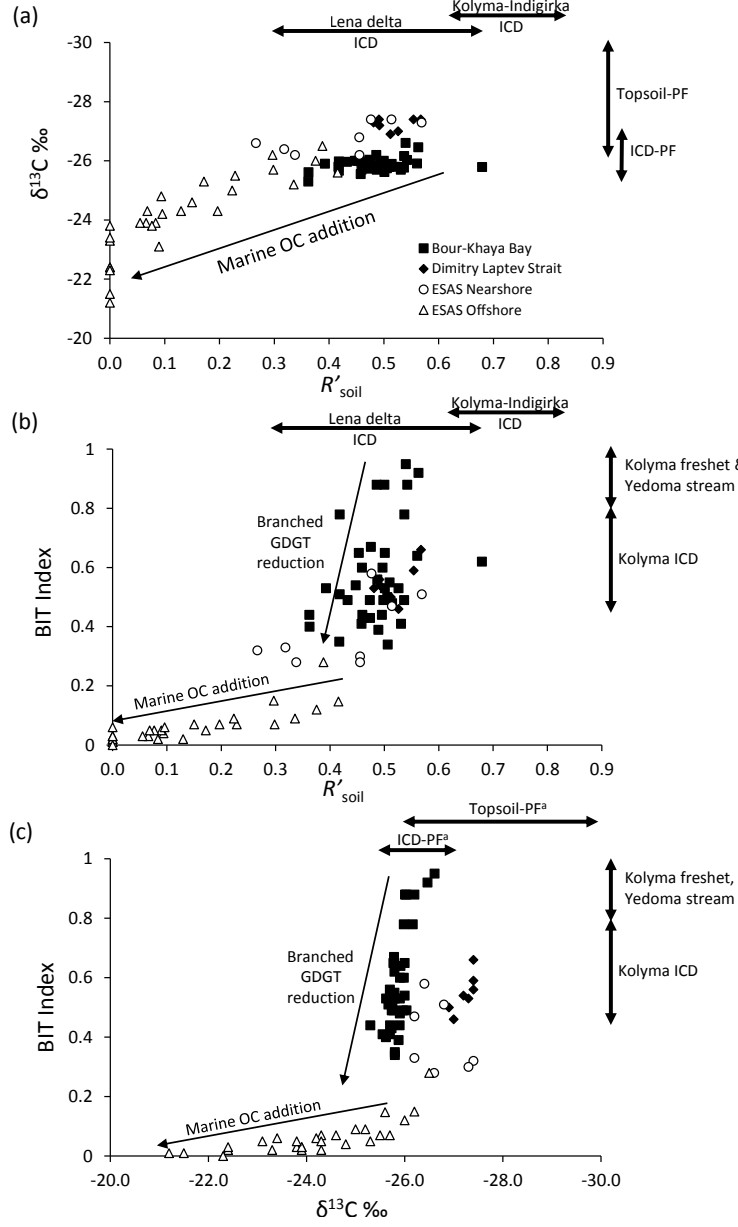

**Figure 5.** Cross plots of $R'_{soil}$ vs. (a) bulk $\delta^{13}$C (Karlsson et al., 2011) and (b) BIT index (Sparkes et al., 2015) and (c) BIT index vs. $\delta^{13}$C in ESAS sediments. Typical values for terrestrial BIT index vs. $\delta^{13}$C endmembers are indicated ($R'_{soil}$ – this study; $\delta^{13}$C – Vonk et al. (2012); BIT index – Bischoff (2013) and Peterse et al. (2014). Note BHP and bulk carbon isotope plot shows linear mixing trend whilst BIT index shows non-linear relationship to both other parameters; the BIT index drops significantly before a shift in isotope ratio to more marine values or shift to lower $R'_{soil}$ values. $R'_{soil}$ endmember values are lower than 1 due to presence of BHT in terrestrial materials (Table S3) and typically lower in Lena River region (mean = 0.50) than in eastern region (mean = 0.76).





**Table 1.** Mean, maximum and minimum values for OM proxy values ($R'_{soil}$, BIT and $\delta^{13}$C) by sample group location and type of material.

| Location | n[a] | $R'_{soil}$ (mean) | $R'_{soil}$ (max) | $R'_{soil}$ (min) | n[a] | BIT (mean) | BIT (max) | BIT (min) | n[a] | $\delta^{13}$C (mean) ‰ | $\delta^{13}$C (max) ‰ | $\delta^{13}$C (min) ‰ |
|---|---|---|---|---|---|---|---|---|---|---|---|---|
| *ISSS-08 Sediments* | | | | | | | | | | | | |
| Bour-Khaya Bay | 47 | 0.49 | 0.80 | 0.34 | 47[b] | 0.58 | 0.95 | 0.26 | 37[c] | -25.9 | -25.3 | -26.6 |
| Dmitry-Laptev Strait | 7 | 0.52 | 0.57 | 0.48 | 7[b] | 0.55 | 0.66 | 0.46 | 7[c] | -27.2 | -26.9 | -27.4 |
| ESAS Nearshore | 9 | 0.41 | 0.57 | 0.27 | 9[b] | 0.35 | 0.58 | 0.10 | 8[c] | -26.8 | -26.2 | -27.4 |
| ESAS Offshore | 29 | 0.14 | 0.42 | 0.00 | 29[b] | 0.06 | 0.28 | 0.00 | 28[c] | -24.2 | -21.2 | -26.5 |
| *Ice Complex* | | | | | | | | | | | | |
| Lena Delta (KUR) | 23 | 0.50 | 0.64 | 0.37 | 23[d] | 0.97 | 1.0 | 0.87. | 23 | n.d.[e] | n.d. | n.d |
| Cape Bykovsky | 1 | 0.68 | | | 1 | n.d. | | | 1 | -25.99 | | |
| Kolyma+Indigirka | 11 | 0.76 | 0.84 | 0.62 | 3[b] | 0.53 | 0.7 | 0.44 | 5 | -24.31 | -23.02 | -25.78 |
| *Literature Data* | | | | | | | | | | | | |
| ICD-Permafrost | | | | | | | | | 374[c] | -26.3±0.67[f] | | |
| Topsoil-Permafrost | | | | | | | | | 20[c] | -28.2±1.98[f] | | |
| Yedoma (Duvannyi Yar) | | | | | 1[g] | 0.82 | | | | | | |
| Yedoma Stream (Duvannyi Yar) | | | | | 8[g] | 0.83 | 0.89 | 0.81 | | | | |
| SPM (Kolyma River) | | | | | 6[g] | | 1.0 | 0.99 | | | | |
| Lena delta permafrost soils | 24[h] | 0.41 | 0.79 | 0.18 | | | | | | | | |

[a] n = number of samples used for calculation of mean for individual parameters;

[b] data from Sparkes et al. (2015);

[c] data from Vonk et al. (2012);

[d] data from Bischoff (2013)

[e] n.d. = not determined;

[f] = mean ± standard deviation;

[g] data from Peterse et al. (2014);

[h] data from Höfle et al. (2015).