# Peer review of "Source, transport and fate of soil organic matter inferred from microbial biomarker lipids on the East Siberian Arctic Shelf"

_Biogeosciences, 2016_

## Referee Comment (RC1) · Anonymous Referee #1 · 23 May 2016

REVIEW COMMENTS

Bischoff et al. present BHP data from samples in the East Siberian Arctic Shelf and adjacent continent to characterize the source and transport of permafrost microbial organic carbon in the high Arctic. The authors compare BHP inventories (BHP abundances and the R'soil proxy) with previously published BIT indices and bulk $\delta$13C data from the same samples. The manuscript focuses a lot on the findings of other studies (e.g. section 3.3) and would benefit from offering additional perspectives to understand the different signals carried by the investigated proxies. For example, the fate of the terrestrial permafrost-derived bacterial OC in the ESAS is still largely unclear and the physico-chemical processes shaping the proxy signals remain under-explored.

[Figure]

The authors conclude that BIT and R'soil represent "different aspects of terrestrial OC" (p. 9, l. 29) and attribute the non-linear correlation of BIT and R'soil to different permafrost sources and transport modes. Based on the large spread of R'soil values in permafrost deposits (see Table 1), however, it is very difficult to assign endmembers. Permafrost soils and ICDs carry extremely heterogeneous BHP/R'soil signatures even within small spatial scales (see Table 1; ICD R'soil 0.37-0.64 for KUR core, soil R'soil 0.18-0.79 for Lena Delta active layer soils – maybe the authors can offer insights as to which factors control the proxy values? For example, do R'soil values show a depth trend in permafrost?). Accordingly, how representative are the mean values for these two endmembers and how can they truly be distinguished? The discussion should be extended to include a dedicated paragraph on this heterogeneity and potential biases arising from it. The heterogeneity should also be considered when discussing W-E trends along the ESAS. Also, the rationale behind the conclusion that the R'soil index carries a more integrated signal of terrestrial OC sources (p. 9, l. 26) including ICDs and riverine BHPs while brGDGTs are only exported fluvially is unclear. Why should the brGDGT inventory of ICDs eroded via coastal erosion not contribute to the sediments while the BHP inventory does? Also, how can the biomarker inventory from ICDs eroded by coastal erosion on the ESAS be distinguished from the biomarker inventory eroded from ICDs further upstream – such as Kurungnakh and other Yedoma delta islands – which is categorized as fluvial OC? These endmembers and potential mixing scenarios should be discussed in more detail in the manuscript. Furthermore, when comparing the proxy-derived pattern of terrestrial OC supply to the ESAS, the authors only discuss potential influences of the continuous vs. discontinuous permafrost catchment coverage. While the Lena catchment also includes discontinuous permafrost areas, these amount to only 20% located entirely in the upstream area. Accordingly, all terrestrial samples investigated in this study derive from continuous permafrost deposits and the change of their R'soil values cannot directly be linked to permafrost conditions. How do/might other factors such as catchment size, runoff, or sediment/POC load shape the proxy signal(s) in the ESAS?

[Figure]

SPECIFIC COMMENTS p. 2, l. 31: replace "that" by "the OC" p. 3, l. 6: correct "reminder" to "remainder" p. 4, l. 1-2: sentence is somewhat repetitive p. 4, l. 3: change "setting" to "settings" p. 4, l. 15-17: add some background to the work of Dogrul-Selver et al., so the sentence better connects with the previous sentence referring to the GDGT and $\delta13C$ study of Sparkes et al. p. 4, l. 24: emphasize that these the exact same samples used by Sparkes et al. p. 5, l. 13: add "the central Lena Delta"; p. 5, l. 19: change capital letter "The" to "the" p. 6, l. 10: were the 1/3 TLE splits (as described in the previous paragraphs; p. 5, l. 32 and p. 6, l. 8) further split into 1/6 and 1/3 fractions? p. 6, l. 14: either use "N2" (p. 6, l. 8) or "nitrogen" consistently p. 6, l. 25: change "concentration" to "concentrations" p. 6, l. 29: change "than" to "that" p. 7, l. 3-4: re-write sentence p. 7, l. 5: change "Concentrations" to "concentrations" p. 7, l. 7: add "their BHP composition" p. 7, l. 10: see comment to Figure 2 (below) p. 7, l. 14: add "in all other nearshore settings" p. 7, l. 18: add "the BHT isomer" p. 8, l. 12: for better comparison add the average percentage of NM soil markers in the Congo estuary p. 8, l. 22: the sentence should rather refer to the general concept of export from terrestrial systems into marine systems; it currently gives the impression that this is only applicable to the Congo deep sea fan. p. 8, l. 23-28: the likelihood of amino-BHP export from wetlands vs. in situ production in sub-sea permafrost could be assessed in more detail. Especially for the Buor Khaya Bay, the amino-BHP concentrations found in the permafrost core from Kurungnakh (this study) and from other studies within the Lena Delta (e.g. Höfle et al., 2015) should allow making better constraints. p. 9, l. 7: change "Eastern" to "eastern" p. 9, l. 15: delete "did" p. 10, l. 6: change "Eastern" to "eastern" p. 10, l. 26: specify "this region" p. 11, l. 12: "Although additional sources from fluvial transport and from material transported via changes in hydrological conduits resulting from thermokarst erosion are also possible" – what type of material is referred to? POC? p. 11, l. 15-16: the Samoylov and Kurungnakh deposits are also genetically different (Holocene fluvial sediment vs. Pleistocene ice complex; Schirrmeister et al., 2011), which is likely mirrored in bulk $\delta13C$ values. p. 11, l. 15-22: $\delta13C$ values should be rounded to one digit p. 11, l. 22: delete "derived" p. 11, l. 15-27: what is the conclusion from this paragraph? p. 11, l. 33-34: add the uncertainty of the % estimate (75±17% and 63±14%) given the large spread of R'soil values in the Kolyma and Indigirka ICDs. Also, these estimates should not refer to OC in total, but only the BHP pool. Other lipid biomarkers or even non-lipid compounds will likely give a very different picture. p. 12, l. 2-3: the estimates of ICD contributions in the ESAS differ largely if the riverine OC is included in the mass balance; for Buor Khaya Bay sediments, Vonk et al. estimate ICD contributions at >50% while Winterfeld et al. (2015; doi:10.5194/bg-12-3769-2015) estimate the ICD contribution to POM around 10%. These differences should be considered. p. 12, l. 4-16: again, considering the large spread of R'soil values within the permafrost samples from each area, and the very limited amount of samples from some areas investigated here (Cape Bykovsky, Kolyma, Indigirka), the trend of increasing R'soil values from W to E is somewhat disputable. Thus, inferences made about the recalcitrance of certain BHPs should be discussed very cautiously. Also, organo-mineral associations seem to be of minor importance in the polygonal tundra (Höfle et al., 2013; doi:10.5194/bg-10-3145-2013) strengthening the argument that the abundance of adenosylhopane and related compounds may as well simply represent a metabolic response to the environmental conditions restricting further side chain elongation (given adenosylhopane is an intermediate in BHP side chain synthesis; Bradley et al., 2010). Figure 1 caption: add "the ISSS 08 expedition"; move brackets for Lantuit et al. citation; Change "KY" to "KI = Kurungnakh Island" Figure 2: panels do not have letters as assigned in the caption and the order seems wrong – panels 1 and 2 should be reversed. Figure 4: Since Figure 4 is referred to in the text when discussing average R'soil indices, it would be helpful to add the mean value contour lines of each group. Figure 5 caption: last sentence – add "in the Lena River region"; specify "eastern region"; add description for abbreviations used (such as PF and superscript a) Table S2: add header to columns K-Z (BHP concentrations)

---

## Referee Comment (RC2) · Anonymous Referee #2 · 7 Jun 2016

This is a review of the paper "Source, transport and fate of soil organic matter inferred from microbial biomarker lipids on the East Siberian Arctic Shelf", by Juliane Bisschof and co-authors, submitted to Biogeosciences as a discussion paper. I have based my review on a paper print version of the manuscript.

The paper is well-written and very well referenced. It discusses an extensive dataset of surface sediments in the East Siberian Arctic, a region that is of interest because of its sensitivity of carbon export in response to past and predicted temperature increase. This dataset is complemented by a number of terrigenous samples.

The results and discussion show that the distribution of bacteriohopanepolyols, expressed as the R'soil, can be used to trace the outflow of terrigenous organic carbon.

[Figure]

It mimicks the stable isotopic signal of the sedimentary organic carbon, which has been used globally to trace terrigenous/marine source of organic matter in riverine outflow systems.

The discussion is centered around two observations: the offset between the BIT and R'soil values, and the east-west gradient in R'soil values in permafrost from west to east.

In the first part of the discussion, I miss a few approaches that would make the discussion more complete. Both the BIT index and R'soil index are determined both by a decrease/increase in terrigenous vs. marine lipids. To compare changes in the degradation of the terrigenous compounds, it is necessary to compare the concentration of these compounds (and possibly also contrasting the different terrigenous BHPs, do they follow the same trend?), rather than comparing the ratios (as these can be influenced by the marine end-member as well).

The offset between brGDGT and BHP behavior is entirely attributed to different spatial distribution of the sources by the authors. The R'soil is proposed to represent a more integrated signature of differential terrigenous sources, including ICD organic matter. However, brGDGTs have been shown to be present in ICD in this study, and in De Jonge et al. (2016). Erosion of ICD OM would thus result in the introduction of both terrigenous GDGT and BHP in the marine environment.

It is also possible that the lipids and bulk parameter represent different pools of OM within the permafrost soils. As mentioned in the manuscript, the lability of the OM depends on the age and organo-mineral interactions. If brGDGTs are more labile to degradation, with BHPs more protected, and bulk OM having the same age/organo-mineral interaction as the BHPs, this can explain the observed offset. Can the authors hypothesize on this? Is there an indication that soil-marker BHPs are preferentially present in older, pre-aged soils, with brGDGTs more abundant in more recently produced material? Can the authors speculate about the size of the particles that BHPs

vs brGDGTs are transported on? Can the study by Tesi et al. (2016) help to explain the observed patterns?

The second part of the discussion, where the increase in permafrost R'soil values is observed in permafrost from west to east, is extensive. Is this shift however also observed in the marine sedimentary R'soil values? It does not seem to be the case based on the mean values, but perhaps the R'soil values in the samples closest to the river mouths follow an east-west trend. If no such a trend is present, the discussion at lines 4-14 (page 12, printed version) is less relevant, as this discusses a trend in properties of sedimentary compounds.

How can the different formation mechanisms for ICD explain the observed east-west gradient in BHP composition? (see fi. L. Schirrmeister, et al., Sedimentary characteristics and origin of the Late Pleistocene Ice Complex on north-east Siberian Arctic coastal lowlands and islands – A review, Quaternary International, Volume 241, Issues 1–2, 2011, Pages 3-25.

Can the authors include the continental OM studied in the title? Fi: "…on the East Siberian Arctic Continent and Shelf".

I have a number of minor corrections below: L5P2, use pool instead of store.

L10P2, have instead of cause

L16P2, Is there a more recent reference for increasing water discharge to the Arctic Ocean?

L31P2, remove 'that'

L6P3, rephrase as 'acting as a positive feedback for climate warming'

L15P3, include 'marine' before crenarchaeol

L24P4, rephrase as: 'with recently published BIT data'

L34P4, refer to the Fig. 1 when discussing the study site

L10P5. Perhaps the zones can be circled in Fig. 1?

L4P7. Perhaps the authors can also summarize the stable isotopic values here?

L25P8. How does the TOC-normalized concentration of aminotetrol and –pentol compare with the values reported in De Jonge et al.(2016)? Can their relative abundance perhaps say something about a terrigenous vs marine source?

L24P10. The BIT values can be significantly different between laboratories, so you have to be careful when comparing values between different studies. See Schouten et al. (2013) An interlaboratory study of TEX86 and BIT analysis of sediments, extracts, and standard mixtures. Geochem. Geophys. Geosyst. 14, 5263–5285.

L7P11. Can the authors include in their discussion whether the relative abundance of the 3 soil-marker BHPs separately is comparable between ICD settings, but also with marine sediments? Does this support an 'unchanged' terrigenous signal to the marine environment (as stated at L30-35P11).

L22P12. What is the pH range studied in the Hofle et al. (2015) paper? Is this pH range relevant for this manuscript?

L5P13: I recommend to use the term 'terrigenous', instead of 'terrestrial'.

L10P15. The journal name is abbreviated.

L29P16. Vol is mentioned twice

L33P16. n/a in the reference should be changed to manuscript number.

L5P18: Journal name is abbreviated

L9P19: subscript in $CO_2$ and $CH_4$.

L22P19: replace n/a

L33P20: replace n/a

L35P20: If this is a book, please include the publisher.

---

## Author Comment (AC1) · 15 Jul 2016

Bart v. Dongen
*Bart.vandongen@manchester.ac.uk*

School of Earth, Atmospheric and
Environmental Sciences
The University of Manchester
Manchester
M13 9PL

Juliane Bischoff
*j.bischoff@hw.ac.uk*

The Lyell Centre
Heriot-Watt University
Edinburgh
EH14 4AS

Response to referee comments of research article ''*Source, transport and fate of soil organic matter inferred from microbial biomarker lipids on the East Siberian Arctic Shelf*'' MS No.: bg-2016-128 submitted for publication in Biogeosciences – Special Issue: Climate–carbon–cryosphere interactions in the East Siberian Arctic Ocean: past, present and future.

Dear Dr Middelburg,

Thank you for accepting the invitation to serve as Associate Editor on our manuscript. In response to the email from Natascha Toepfer, dated 14/06/16, advising us that the open discussion of our manuscript has been closed, we are pleased to provide you with a response to the 2 referee comments that we received. For clarity, referee comments were broken down in numbered sections and are written in italics, followed by our corresponding response.

The procedure stated in your mail advises us to provide a response without submitting a revised manuscript. If changes in the manuscript were suggested by the referees, we have included either excerpts of the suggested revised manuscript in our response or indicated where and how the reviewer's comments will be incorporated in the revised manuscript.

We hope that you will find our approach suitable and beneficial in assessing our manuscript for publication in Biogeosciences. If you require any further information, please do not hesitate to contact us.

Bart van Dongen & Juliane Bischoff (in the name of all the co-authors)

1. *Bischoff et al. present BHP data from samples in the East Siberian Arctic Shelf and adjacent continent to characterize the source and transport of permafrost microbial organic carbon in the high Arctic. The authors compare BHP inventories (BHP abundances and the R'soil proxy) with previously published BIT indices and bulk δ13C data from the same samples. The manuscript focuses a lot on the findings of other studies (e.g. section 3.3) and would benefit from offering additional perspectives to understand the different signals carried by the investigated proxies. For example, the fate of the terrestrial permafrost-derived bacterial OC in the ESAS is still largely unclear and the physico-chemical processes shaping the proxy signals remain under-explored.*

   **Response:** We thank reviewer #1 for her/his comments on our manuscript and agree with the brief summary of our manuscript. However, we are not entirely clear about the meaning of the reviewers comment that 'the fate of terrestrial permafrost derived bacterial OC **in the ESAS** is still largely unclear'. We disagree with this as we have previously shown using other microbial biomarkers (branched-GDGTs; Sparkes et al., 2015; doi:10.5194/bg-12-3753-2015) that these compounds that are primarily transported via fluvial mechanisms (with much lower concentrations from ice complex deposits; see response to points 3, 9, 35 and 45 below), are rapidly degraded on the ESAS, mainly near the point of entry (in this case mainly the river mouths). However, we do agree that there is much more to learn about the sources of this material and in this manuscript we have only just begun to explore the wide diversity of potential source material and to consider the 'physico-chemical' factors affecting the proxy signals. We do for example already point out that recent studies have shown that pH appears to have a profound effect on terrestrial BHP distributions (Höfle et al., 2015; doi: 10.1016/j.orggeochem.2015.08.002; original MS page 12, lines 21-22). This effect is further explored in a recently published manuscript (Talbot et al., 2016; doi: 10.1016/j.orggeochem.2016.04.01), which will be incorporated into the revised manuscript describing the competing effects of pH and temperature. The former leads to very low proportions of soil-marker BHPs (as seen in *sphagnum* peat) whilst cold temperatures can lead to much higher relative proportions of soil markers. These effects are also further expanded upon in the response to points 2 and 37 below.

2. *The authors conclude that BIT and R'soil represent "different aspects of terrestrial OC" (p. 9, l. 29) and attribute the non-linear correlation of BIT and R'soil to different permafrost sources and transport modes. Based on the large spread of R'soil values in permafrost deposits (see Table 1), however, it is very difficult to assign endmembers. Permafrost soils and ICDs carry extremely heterogeneous BHP/R'soil signatures even within small spatial scales (see Table 1; ICD R'soil 0.37-0.64 for KUR core, soil R'soil 0.18-0.79 for Lena Delta active layer soils – maybe the authors can offer insights as to which factors control the proxy values? For example, do R'soil values show a depth trend in permafrost?). Accordingly, how representative are the mean values for these two endmembers and how can they truly be distinguished? The discussion should be extended to include a dedicated paragraph on this heterogeneity and potential biases arising from it. The heterogeneity should also be considered when discussing W-E trends along the ESAS.*

   **Response:** We agree that permafrost deposits, their BHP signatures and associated $R'_{soil}$ values presented here are heterogeneous, which is one of the key messages of our manuscript (see original MS page 13, lines 12 to 17). We also agree that 'it is difficult to assign endmembers'. However, there are some strong consistencies. For example, the mean $R'_{soil}$ value for the KUR

core is 0.5 (Table 1) and comparable to the mean value for the Buor Khaya Bay sediments ($R'_{soil}$ 0.5) as can be clearly seen in Figure 4. In contrast, the active layer soil samples from other nearby locations in the Lena delta have a wider range of values but represent only a shallow depth range (maximum depth 48 cm; Höfle et al., 2015). This might suggest that the deeper samples from the KUR core more closely represent the wider sources in the area including points of coastal erosion where deeper profiles are affected rather than only that material that is mobilised via the annual thaw cycle and subsequent movement of water through the active layer. Indeed we already point out (original MS page 10, lines 31-33) that "this simple average will likely not represent a spatially and depth resolved average for the region"; clearly a more comprehensive survey of the different sources (including different age/depth and additional settings including river transported material) is required to better assess endmembers. This is one of the main reasons why we suggest not to focus exclusively on a single mean value but instead use the full range of available values for terrestrial source materials. This is to show that the range is broadly consistent with those found in Buor-Khaya Bay and other nearshore ESAS sediments. In fact, for the same reasons we only make limited use (no mean endmembers) of these values to come up with any kind of carbon budget (original MS page 11 line 33 to page 12 line3). See also our response to comment 35.

At this time some environmental factors that can affect the $R'_{soil}$ proxy value are emerging and include pH which leads to lower relative levels of soil-marker BHPs (Höfle et al., 2015; Talbot et al., 2016); temperature which can lead to higher values although with significant variations likely due to additional factors including but not limited to pH (Talbot et al., 2016) and undoubtedly other physiological factors (e.g. salinity) affecting the source microbiological community. We observed no clear depth trends across the profiles studied here with some having higher $R'_{soil}$ values at depth and others at surface or intermediate levels. This is also true for the data from Höfle et al. (2015).

Regarding the east-west trend (in $R'_{soil}$ value for the ice complex samples) we would like to clarify that we do not imply a gradual progression from lower values in the west to higher values in the east. Rather we identify extremes noting that the very highest values (>0.7) are only found in samples from the easternmost settings in the Kolyma and Indigirka areas with lower values more common in the Lena area. This will be further clarified in the revised manuscript together with some possible reasons for this (colder/dryer conditions in the East) as well as depositional age and formation mechanism (see also point 49) which are already discussed at page 12, lines 4-7 (Original MS). Further clarification is given below in our response to points 4, 37 and 49.

3. *Also, the rationale behind the conclusion that the R'soil index carries a more integrated signal of terrestrial OC sources (p. 9, l. 26) including ICDs and riverine BHPs while brGDGTs are only exported fluvially is unclear. Why should the brGDGT inventory of ICDs eroded via coastal erosion not contribute to the sediments while the BHP inventory does? Also, how can the biomarker inventory from ICDs eroded by coastal erosion on the ESAS be distinguished from the biomarker inventory eroded from ICDs further upstream – such as Kurungnakh and other Yedoma delta islands – which is categorized as fluvial OC? These endmembers and potential mixing scenarios should be discussed in more detail in the manuscript.*

   **Response:** The potential origins of GDGTs in this region has already been discussed in detail in a previous paper by Sparkes et al. (2015). In short, the main reason we consider GDGTs to be primarily delivered to the ESAS transported by fluvial transport is because they are only present in low to very low abundances in the ICD, when compared to the riverine transported material. Therefore, coastal erosion of ICD cannot be a major source of branched GDGTs to the ESAS (see

for instance the model presented in Sparkes et al. 2015, which estimated that ICD erosion supplied 77% of the sediment on the ESAS, 44% of the OC but only 21% of the br-GDGTs). We agree that material from the coastal erosion cannot be distinguished from material eroded from ICDs further upstream and subsequently transported by river, but in the case of the GDGTs, the substantial contribution from topsoil erosion and riverine in-situ produced GDGTs will likely overwhelm this low contribution from ICDs (Peterse et al., 2014; doi: 10.1002/2014JG002639). This is in contrast to the BHPs, which, as we show in this MS, can be significant components in ICDs. This suggests, in contrast to the GDGTs, a much more mixed contribution to the shelf, as supported by the correlation between $R'_{soil}$ and the bulk stable carbon isotope values (Fig. 5a) and the fact that De Jonge et al. (2016) highlighted that BHPs can indeed be transported in suspended particulate matter from Arctic rivers. However, it is currently not possible to disentangle the contributions to this much more integrated signal of riverine input, upstream and coastal erosion of ICD.

The current, limited dataset of terrestrial endmember materials is not appropriate to discuss 'these endmembers and potential mixing scenarios' as suggested by the reviewer and we will refrain from doing so. As pointed out by this reviewer and by ourselves in the manuscript, we are aware about the stark heterogeneity of different permafrost types, sites and depths and the associated issues in constraining endmembers.

4. *Furthermore, when comparing the proxy-derived pattern of terrestrial OC supply to the ESAS, the authors only discuss potential influences of the continuous vs. discontinuous permafrost catchment coverage. While the Lena catchment also includes discontinuous permafrost areas, these amount to only 20% located entirely in the upstream area. Accordingly, all terrestrial samples investigated in this study derive from continuous permafrost deposits and the change of their R'soil values cannot directly be linked to permafrost conditions. How do/might other factors such as catchment size, runoff, or sediment/POC load shape the proxy signal(s) in the ESAS?*

   **Response:** We agree that our manuscript links the $R'_{soil}$ to the distribution of continuous/discontinuous permafrost and might convey the impression that the terrestrial endmember of $R'_{soil}$ is controlled by a transition of continuous/discontinuous permafrost from East to West. In the revised manuscript we will highlight that we are talking about two separate regional provinces, both underlain by mostly continuous permafrost, but with samples that are subject to different (1) environmental conditions (e.g. drier and colder conditions in the east), (2) ages and (3) formation mechanism of the permafrost. The latter two points were also highlighted by the other reviewer (comment number 49).

   We agree with this reviewer that the environmental controls of $R'_{soil}$ are currently not sufficiently constrained. This is indeed very interesting and necessary, but falls outside the scope of the current study and warrants further, designated investigation. As mentioned before, we recognize the limitations of the sample set in this study and have highlighted the stark heterogeneities (response to comments 1 and 2 above). To constrain the effect of 'catchment size, run off, or POC load' a much more substantial set of samples in different spatial and temporal resolutions and further thorough investigation would be required.

*SPECIFIC COMMENTS*
5. *p. 2, l. 31: replace "that" by "the OC"*

**Response:** We suggest the sentence be revised as follows: "Additionally, OC is stored, frozen, within coastal ice complex deposits (ICD) – this can also be a major source of terrOC to the Arctic Ocean…"

6. *p. 3, l. 6: correct "reminder" to "remainder"*

   **Response:** This will be corrected in the revised manuscript.

7. *p. 4, l. 1-2: sentence is somewhat repetitive*

   **Response:** We will introduce the proxy earlier in the MS and delete this sentence.

8. *p. 4, l. 3: change "setting" to "settings"*

   **Response:** This will be corrected in the revised manuscript.

9. *p. 4, l. 15-17: add some background to the work of DogrulSelver et al., so the sentence better connects with the previous sentence referring to the GDGT and δ13C study of Sparkes et al.*

   **Response:** We agree with the reviewer and suggest to add the following paragraph that expands on the study of Doğrul Selver et al., 2015 (starting at page 4 line 13, original MS):
   ……Recently, Sparkes et al. (2015) used the GDGT based BIT proxy to trace terrestrial OM in the same samples on the ESAS shelf and found a decoupling between BIT and bulk $\delta^{13}$C, suggesting that GDGTs were (primarily) sourced via riverine transport and not from erosion of coastal ICD. In addition, a strong linear correlation between bulk $\delta^{13}$C and $R'_{soil}$ and a strong but non-linear relationship between the BIT index and $R'_{soil}$ was observed by Doğrul Selver et al. (2015) in surface sediments along the offshore transect off the Kolyma River. This suggest a decoupling between these microbial based biomarker proxies and different and/or additional sources of BHPs to the ESAS compared to the GDGTs…..

10. *p. 4, l. 24: emphasize that these the exact same samples used by Sparkes et al.*

    **Response:** To clarify that Sparkes et al. (2015) investigated the same samples we suggest to add 'in the same samples' in to page 4, line 24 of the original manuscript. We also suggest to emphasise this point by adding 'of the same sites' to page 4, line 13 of the original manuscript.

11. *p. 5, l. 13: add "the central Lena Delta";*

    **Response:** This will be corrected in the revised manuscript.

12. *p. 5, l. 19: change capital letter "The" to "the"*

    **Response:** This will be corrected in the revised manuscript.

13. *p. 6, l. 10: were the 1/3 TLE splits (as described in the previous paragraphs; p. 5, l. 32 and p. 6, l. 8) further split into 1/6 and 1/3 fractions?*

    **Response**: No, the 'aliquots (1/6 for ESAS sediments and 1/3 for terrestrial materials) of the TLE' refer to the initial amount of sample that was extracted. We will address this in the manuscript by stating that 1/6th was used for BHP analysis on page 5 line 34 of the original manuscript.

*14. p. 6, l. 14: either use "N2" (p. 6, l. 8) or "nitrogen" consistently*

**Response:** We checked the consistent use of $N_2$/nitrogen throughout the document and found two occasions where it was used inconsistently. This will be corrected in the revised manuscript.

*15. p. 6, l. 25: change "concentration" to "concentrations"*

**Response:** This will be corrected in the revised manuscript.

*16. p. 6, l. 29: change "than" to "that"*

**Response:** This will be corrected in the revised manuscript.

*17. p. 7, l. 3-4: re-write sentence*

**Response:** This will be corrected in the revised manuscript by removing the word "are" from page 7, line 4.

*18. p. 7, l. 5: change "Concentrations" to "concentrations"*

**Response:** This will be corrected in the revised manuscript

*19. p. 7, l. 7: add "their BHP composition"*

**Response:** This will be corrected in the revised manuscript

*20. p. 7, l. 10: see comment to Figure 2 (below)*

**Response:** *See response to comment 39 on Figure 2 below*

*21. p. 7, l. 14: add "in all other nearshore settings"*

**Response:** This will be corrected in the revised manuscript

*22. p. 7, l. 18: add "the BHT isomer"*

**Response:** This will be modified in the revised manuscript.

*23. p. 8, l. 12: for better comparison add the average percentage of NM soil markers in the Congo estuary*

**Response:** The study from Spencer-Jones et al. (2015) included only one sample from the Congo estuary, which solely contained Adenosylhopane.

*24. p. 8, l. 22: the sentence should rather refer to the general concept of export from terrestrial systems into marine systems; it currently gives the impression that this is only applicable to the Congo deep sea fan.*

**Response:** We agree and will delete 'from the Congo' in the revised manuscript.

*25. p. 8, l. 23-28: the likelihood of amino-BHP export from wetlands vs. in situ production in sub-sea permafrost could be assessed in more detail. Especially for the Buor Khaya Bay, the amino-BHP concentrations found in the permafrost core from Kurungnakh (this study) and from other studies within the Lena Delta (e.g. Höfle et al., 2015) should allow making better constraints.*

**Response:** The previous study of Lena delta active layer soils (Höfle et al., 2015) found very low levels of amino-BHPs indicative of aerobic methane oxidation almost exclusively in polygon centres and primarily in the shallow organic horizons where they represented between 0 and 3.3% of the total BHP assemblage. Similarly, in the KUR core we also found very low levels of the aminoBHPs at some points down core (data not shown). Whilst this indicates that transport of these compounds from ice complex might be possible, levels are too low and inconsistent to make any conclusive assessment at this time. However, we also recognise the significance of the transport of POC from thermokarst settings (Vonk et al., 2015) and consider it possible that aminoBHPs may be produced by methanotrophs in such settings (e.g. methane saturated thermokarst lakes) so there is potential for these settings to be source areas (cf. tropical wetlands as sources of aminoBHPs to tropical deep sea-fans; Talbot et al., 2014; Wagner et al., 2014; Spencer-Jones et al., 2015). Data is currently lacking to consider this further. Conversely, although there is evidence of methane oxidation occurring in the water column and sediments of the ESAS at sites of sub-sea permafrost decomposition (see references from Shakhova et al. in the original MS), it is unknown if this is mediated by the type of methanotrophs that biosynthesise the biomarkers discussed here (see review in Talbot et al., 2014; doi: 10.1016/j.gca.2014.02.035). We are therefore confident that our cautious approach in which we recognise some possibilities but do not speculate further is fully appropriate and choose not make any additional changes to the text.

*26. p. 9, l. 7: change "Eastern" to "eastern"*

**Response:** This will be corrected in the revised manuscript.

*27. p. 9, l. 15: delete "did"*

**Response:** We have carefully checked the sentence and suggest against deleting 'did'.

*28. p. 10, l. 6: change "Eastern" to "eastern"*

**Response:** This will be corrected in the revised manuscript.

*29. p. 10, l. 26: specify "this region"*

**Response:** This will be rewritten in the revised manuscript to make clear that this is the (East Siberian) Arctic region.

*30. p. 11, l. 12: "Although additional sources from fluvial transport and from material transported via changes in hydrological conduits resulting from thermokarst erosion are also possible" – what type of material is referred to? POC?*

**Response:** Yes, we refer to POC (Section 2.2.3 in Vonk et al., 2015) and this will be clarified in the revised manuscript.

*31. p. 11, l. 15-16: the Samoylov and Kurungnakh deposits are also genetically different (Holocene fluvial sediment vs. Pleistocene ice complex; Schirrmeister et al., 2011), which is likely mirrored in bulk δ13C values.*

**Response:** We thank the reviewer for pointing this out and will change the sentence to: ….and may be because these deposits are genetically different (Holocene fluvial sediment vs. Pleistocene ICD) or reflect input from aquatic…..

*32. p. 11, l. 15-22: δ13C values should be rounded to one digit*

**Response:** We agree with the reviewer and will consistently present the $\delta^{13}$C values in the revised manuscript with one digit.

*33. p. 11, l. 22: delete "derived"*

**Response:** This will be modified in the revised manuscript.

*34. p. 11, l. 15-27: what is the conclusion from this paragraph?*

**Response:** This paragraph was intended to lead into the %OC estimates based on BHPs. However, we agree with this reviewer that this is not completely clear and suggest adding the following sentence to the end of this paragraph : 'These terrestrial OM estimates can now be compared to estimates based on the BHP concentrations/compositions obtained using a similar approach with the results from the ICD samples as the terrestrial endmembers.'
In addition we also realise that section 3.3 is not solely about terrestrial endmembers anymore and suggest changing the heading of this section to 'terrestrial endmembers and implications for ESAS sedimentary carbon budgets'

*35. p. 11, l. 33-34: add the uncertainty of the % estimate (75±17% and 63±14%) given the large spread of R'soil values in the Kolyma and Indigirka ICDs. Also, these estimates should not refer to OC in total, but only the BHP pool. Other lipid biomarkers or even non-lipid compounds will likely give a very different picture.*

**Response:** Our study shows that the BHPs based $R'_{soil}$ is clearly, linearly correlated with the bulk carbon isotope values (r$^2$=-0.73, p<0.001), indicating that the $R'_{soil}$ is tracing a more integrated signature i.e. the bulk organic carbon. This close match between $\delta^{13}$C and $R'_{soil}$ was also shown by Doğrul Selver et al. (r$^2$ = 0.96; 2015). We are therefore confident presenting estimations on the %OC based on $R'_{soil}$ for the ESAS. However, as already highlighted in response to comment 2, we suggest to consider the range in terrestrial endmembers rather than focusing on a single mean value and therefore suggest not to add the uncertainty of the % estimate but rather report the full range based on the range of endmembers (58 to 92% and 49 to 77%) in the revised manuscript. Clearly improved data would strengthen these estimates and future field expeditions should collect year-round river samples to complement the growing ICD dataset (see also response to comment 36 below). We agree that other lipid markers will likely give another picture. Sparkes et al. (2015) show a non-linear correlation between $\delta^{13}$C and BIT, indicating that the BIT index does not trace bulk material, but is driven by a rapid reduction in brGDGTs and subsequent slow marine production and therefore traces more labile riverine material that does not represent the bulk OC pool. Therefore, estimation of %OC based on the GDGT based BIT index might not be representative for the whole OC pool and should be treated with caution.

36. *p. 12, l. 2-3: the estimates of ICD contributions in the ESAS differ largely if the riverine OC is included in the mass balance; for Buor Khaya Bay sediments, Vonk et al. estimate ICD contributions at >50% while Winterfeld et al. (2015; doi:10.5194/bg-12-3769-2015) estimate the ICD contribution to POM around 10%. These differences should be considered.*

> **Response:** We apologize since we feel that we have confused this reviewer by comparing our estimated to the % derived from ICD. This was a mistake and should have been % derived from terrestrial OM (e.g. % ICD and topsoil). This means that although the reviewer is correct that there is a difference between estimated ICD contributions based on Vonk et al. and Winterfeld et al. this is not relevant for our calculations, considering that we can (currently) not distinguish between soil derived and ICD derived OM using BHPs. We suggest changing the last part of this section to "......These values should be treated with caution given the limited number of samples involved, but are close to the average value reported for the ESAS surface sediments based on dual-carbon-isotope ($\delta$ $^{13}$C and $\Delta^{14}$C) mixing models (74±14% from terrestrial origin; Vonk et al., 2012). This increase in $R'_{soil}$ ....."

37. *p. 12, l. 4-16: again, considering the large spread f R'soil values within the permafrost samples from each area, and the very limited amount of samples from some areas investigated here (CapeBykovsky, Kolyma, Indigirka), the trend of increasing R'soil values from W to E is somewhat disputable. Thus, inferences made about the recalcitrance of certain BHPs should be discussed very cautiously. Also, organo-mineral associations seem to be of minor importance in the polygonal tundra (Höfle et al., 2013; doi:10.5194/bg-10-31452013) strengthening the argument that the abundance of adenosylhopane and related compounds may as well simply represent a metabolic response to the environmental conditions restricting further side chain elongation (given adenosylhopane is an intermediate in BHP side chain synthesis; Bradley et al., 2010).*

> **Response:** We agree with this reviewer about the limitation in the amount and spatial distribution of our samples. As mentioned previously, we will modify the revised manuscript carefully to avoid implying a transition/trend from West to East (see our response to comment 2 and 4 on this issue). We have checked the reference suggested by the reviewer relating to the organo-mineral association and will include it in the revised manuscript. We suggest adding the following text at P 12, line 13 of the original manuscript: "Adenosylhopane, the most abundant single soil marker BHP on the ESAS, is the only BHP containing an aromatic moiety (adenine). Although Höfle et al. (2013) found organo-mineral associations to be of minor importance in the polygonal tundra of the Lean delta, organic-mineral interactions may still be among several factors explaining the high relative abundance of these compounds under certain conditions."

38. *Figure 1 caption: add "the ISSS 08 expedition"; move brackets for Lantuit et al. citation; Change "KY" to "KI = Kurungnakh Island"*

> **Response**: We will modify the Figure 1 caption following the suggestions of the reviewer.

39. *Figure 2: panels do not have letters as assigned in the caption and the order seems wrong – panels 1 and 2 should be reversed.*

> **Response:** We will submit an updated version of this figure with the revised manuscript.

40. *Figure 4: Since Figure 4 is referred to in the text when discussing average R'soil indices, it would be helpful to add the mean value contour lines of each group.*

**Response:** The information on the average $R'_{soil}$ values at the different locations is already included in Table 1. We have tried adding the contour lines as suggested by the reviewer, but found them rather confusing in an already busy plot. If this is a concern for the editor we are happy to adapt the figure as suggested.

*41. Figure 5 caption: last sentence – add "in the Lena River region"; specify "eastern region"; add description for abbreviations used (such as PF and superscript a)*

**Response:** We will add 'in the Lena River region' and 'the eastern region' as well as the description for abbreviation PF (*PF = Permafrost*) to the figure caption and remove superscript [a] as it is no longer required and the information on literature sources is contained in the figure caption.

*42. Table S2: add header to columns K-Z (BHP concentrations)*

**Response:** The column header will be added in the revised manuscript

Response to Reviewer 2:

*43. This is a review of the paper "Source, transport and fate of soil organic matter inferred from microbial biomarker lipids on the East Siberian Arctic Shelf", by Juliane Bisschof and co-authors, submitted to Biogeosciences as a discussion paper. I have based my review on a paper print version of the manuscript. The paper is well-written and very well referenced. It discusses an extensive dataset of surface sediments in the East Siberian Arctic, a region that is of interest because of its sensitivity of carbon export in response to past and predicted temperature increase. This dataset is complemented by a number of terrigenous samples. The results and discussion show that the distribution of bacteriohopanepolyols, expressed as the R'soil, can be used to trace the outflow of terrigenous organic carbon. It mimics the stable isotopic signal of the sedimentary organic carbon, which has been used globally to trace terrigenous/marine source of organic matter in riverine outflow systems. The discussion is centered around two observations: the offset between the BIT and R'soil values, and the east-west gradient in R'soil values in permafrost from west to east.*

**Response:** We thank this reviewer for his/her interest in our research and his/her kind and constructive comments on our manuscript. We agree with the summary of our manuscript, as understood by the reviewer, with the exception of the last point made by the reviewer ('*the east-west gradient in R'soil values in permafrost from west to east'*). We were made aware by both reviewers that the current manuscript conveys the impression of a transition/trend from east to west (see also our response to comment 2 and 4 of the previous reviewer). Unfortunately, due to the limitations of our sample set, we are not able to address a transect and/or transitional gradient. We will carefully modify the revised manuscript as outlined in our response addressing comments 2 and 4 by the previous reviewer.

*44. In the first part of the discussion, I miss a few approaches that would make the discussion more complete. Both the BIT index and R'soil index are determined both by a decrease/increase in terrigenous vs. marine lipids. To compare changes in the degradation of the terrigenous compounds, it is necessary to compare the concentration of these compounds (and possibly also contrasting the different terrigenous BHPs, do they follow the same trend?), rather than comparing the ratios (as these can be influenced by the marine end-member as well).*

**Response:** Generally speaking, the amounts in μg/g OC of both terrestrial GDGTs and BHP soil markers are the highest near shore and decline with distance from the mainland. This is presented in Figure 3b in this manuscript as well as in the corresponding figure addressing brGDGTs in Sparkes et al. (2015). With respect to the behaviour of different terrigenous BHPs, we can confirm that these follow similar trends off shore. We suggest changing the text on page 7, line 27 of the original manuscript to make this clear to ''……the concentrations of all non-methylated soil markers are highest in samples closer to the coast (0-100 km) and decrease with distance from the river outflows (Figures 2b and 3b; Table S2) showing similar trends…''.

45. *The offset between brGDGT and BHP behavior is entirely attributed to different spatial distribution of the sources by the authors. The R'soil is proposed to represent a more integrated signature of differential terrigenous sources, including ICD organic matter. However, brGDGTs have been shown to be present in ICD in this study, and in De Jonge et al. (2016). Erosion of ICD OM would thus result in the introduction of both terrigenous GDGT and BHP in the marine environment. It is also possible that the lipids and bulk parameter represent different pools of OM within the permafrost soils. As mentioned in the manuscript, the lability of the OM depends on the age and organo-mineral interactions. If brGDGTs are more labile to degradation, with BHPs more protected, and bulk OM having the same age/organomineral interaction as the BHPs, this can explain the observed offset. Can the authors hypothesize on this?*

    **Response:** We agree that the offset between the behaviour of GDGTs and BHPs on the shelf and their respective OM proxies is discussed with respect to their sources. However, whilst we agree that brGDGTs are present in ICD organic matter, it was shown that (1) the amounts of brGDGTs are low to very low (Sparkes et al. 2015) and (2) a compositional offset between the brGDGTs in the coastal ICD compared to the Yenisej river mouth was observed (de Jonge 2015, GCA). Therefore, the erosion of coastal ICD only results in a minor contribution to the amounts of brGDGTs found on the Artic Shelf compared to riverine produced GDGTs (Sparkes et al. 2015, de Jonge et al., 2015). For more detail please see our response to comment 3 of the other reviewer. Also, given the information provided in comment 37 from the other reviewer, we prefer to refrain for speculating further regarding the potential for organo-mineral interaction until more data is available.

46. *Is there an indication that soil-marker BHPs are preferentially present in older, pre-aged soils, with brGDGTs more abundant in more recently produced material?*

    **Response:** This is an interesting suggestion. However, we are not in a position to address this comment with the current set off samples and thus falls outside the scope of this manuscript.

47. *Can the authors speculate about the size of the particles that BHPs vs brGDGTs are transported on? Can the study by Tesi et al. (2016) help to explain the observed patterns?*

    **Response:** Tesi et al. (2016) reported that "OC is mainly associated with fine/ultrafine mineral particles" although "biomarkers indicate that the selective transport of fine-grained sediment results in mobilizing high-molecular weight (HMW) lipid-rich, diagenetically altered TerrOC while lignin-rich, less degraded TerrOC is retained near the coast". We already speculate about the BHPs (page 12, line 11-13), however, the reviewer asks about BHPs vs. GDGTs. Neither of these were included in the suite of compounds investigated by Tesi et al. (2016). We therefore feel this would be taking the speculation too far and suggest that this should be the subject of future study.

*48. The second part of the discussion, where the increase in permafrost R'soil values is observed in permafrost from west to east, is extensive. Is this shift however also observed in the marine sedimentary R'soil values? It does not seem to be the case based on the mean values, but perhaps the R'soil values in the samples closest to the river mouths follow an east-west trend. If no such a trend is present, the discussion at lines 4-14 (page 12, printed version) is less relevant, as this discusses a trend in properties of sedimentary compounds.*

> **Response:** We would like to emphasise again that we are not considering a west to east transition or trend (see responses to comments 2 and 4, previous reviewer and 43, this reviewer) but rather regional provinces and extremes. Further, the samples closest to the river mouths are not directly comparable between Kolyma and Bour-Khaya Bay. The first sample from the Kolyma river mouth already has a 160 km distance from the river mouth, compared to 25 km in Bour Khaya Bay. Therefore, we feel we cannot make any statements on the discrepancies/comparison of the marine sedimentary $R'_{soil}$ values.

*49. How can the different formation mechanisms for ICD explain the observed east-west gradient in BHP composition? (see fi. L. Schirrmeister, et al., Sedimentary characteristics and origin of the Late Pleistocene Ice Complex on north-east Siberian Arctic coastal lowlands and islands – A review, Quaternary International, Volume 241, Issues 1–2, 2011, Pages 3-25. Can the authors include the continental OM studied in the title? Fi: "...on the East Siberian Arctic Continent and Shelf".*

> **Response:** First we refer to our response to points 2 and 4 of the other reviewer as well as point 43 and emphasise that we did not intend to imply a gradient. Given the limited data currently available for ICD we can only identify extremes at this time. We agree with the reviewer that different formation mechanisms of permafrost in East and West will likely have an effect on the suite of BHPs and also note that regional environmental conditions (i.e. colder, drier conditions in the East [see response to point 4 from the other Reviewer]) and age of the permafrost deposits will likely all be contributing factors. Unfortunately, it remains unclear how this would affect the BHP composition and warrants further study. We will clarify these factors in the revised version (after line 4-7, page 12 – see suggested changes to comment 2).
>
> Concerning the suggested change in the title: We do not think that adding ''Continent'' to the title is appropriate as we are focusing on the behaviour of OM on the shelf. We are only using the much smaller number of samples from the land as endmembers to assess what happens to the material in the marine system.

*I have a number of minor corrections below:*
*50. L5P2, use pool instead of store.*

> **Response:** This will be modified in the revised manuscript.

*51. L10P2, have instead of cause*

> **Response:** This will be modified in the revised manuscript.

*52. L16P2, Is there a more recent reference for increasing water discharge to the Arctic Ocean?*

**Response:** We will add the reference Rawlings et al. (2010; below) which states for example "Pan-Arctic river discharge….has also risen over recent decades…..the-long term increase in river discharge from large Eurasian rivers is perhaps the most consistent trend evidencing Arctic FWC [freshwater circulation] intensification." However, one of the primary references cited in this work is Peterson et al. (2002) therefore we will retain this reference as well.

Rawlings, M.A., Steele, M., Holland, M.M., Adam, J.C., Cherry, J.E., Francis, J.A., et al.: Analysis of the Arctic system for freshwater cycle intensification: Observations and expectations, Journal of Climate. 23, 5715-5736, doi: 10.1175/2010JCLI3421.1

*53. L31P2, remove 'that'*

**Response:** This sentence was also commented on by the previous reviewer and will be modified in the revised manuscript.

*54. L6P3, rephrase as 'acting as a positive feedback for climate warming'*

**Response:** This will be modified in the revised manuscript.

*55. L15P3, include 'marine' before crenarchaeol*

**Response:** This will be modified in the revised manuscript.

*56. L24P4, rephrase as: 'with recently published BIT data'*

**Response:** This will be modified in the revised manuscript.

*57. L34P4, refer to the Fig. 1 when discussing the study site*

**Response:** This will be modified in the revised manuscript.

*58. L10P5. Perhaps the zones can be circled in Fig. 1?*

**Response:** We will modify the figure in such a way that the zones are clear.

*59. L4P7. Perhaps the authors can also summarize the stable isotopic values here?*

**Response:** We will summarize the stable isotope values briefly in this section in the revised version of the manuscript.

*60. L25P8. How does the TOC-normalized concentration of aminotetrol and –pentol compare with the values reported in De Jonge et al.(2016)? Can their relative abundance perhaps say something about a terrigenous vs marine source?*

**Response:** First we stress that the Yenisei River and outflow environment as used in the study reported by de Jonge et al. (2016) is a significantly different environment if compared to East Siberian Arctic region (e.g. drainage basins of the Lena, Indigirka and Kolyma) making a direct comparison between both studies difficult. Furthermore, we focus on aminopentol in our response as it has the most constrained biological source unlike aminotetrol which can be found

in organisms other than methanotrophs (see review in Talbot et al., 2014). The highest values for TOC normalised concentrations of aminopentol (**1c**; Table S2) in ESAS sediments were found in Buor Khaya Bay (0 - 9.5 µg $g_{OC}^{-1}$) and it was only rarely present and only at or below 5.7 µg $g_{OC}^{-1}$ in the other areas (Table S2). Even the highest value is considerably lower than the highest values found in the sediments in the De Jonge et al. (2016) study (0 – 48 µg $g_{OC}^{-1}$) so the settings do not appear to be comparable. We therefore reiterate our response to point 25 from the other Reviewer in that concentrations are too low and data is currently lacking from a range of endmembers (e.g. thermokarst lakes) for this region to make any further assessment at this time.

We suggest to add a line to the manuscript at page 8, line 25: "Here in Buor-Khaya Bay, concentrations of aminopentol (**1c**) ranged from 0 – 9.5 µg $g_{OC}^{-1}$, considerably lower than values reported in De Jonge et al., (2916; 0 – 48 µg $g_{OC}^{-1}$. It is possible therefore that the aminopentol signature is either fluvially transported from areas where aerobic methane has taken place within the catchment (i.e. wetlands, lakes) or alternatively ……."

61. *L24P10. The BIT values can be significantly different between laboratories, so you have to be careful when comparing values between different studies. See Schouten et al. (2013) An interlaboratory study of TEX86 and BIT analysis of sediments, extracts, and standard mixtures. Geochem. Geophys. Geosyst. 14, 5263–5285.*

> **Response:** Generally, we agree with the reviewer's comment, although this is not strictly part of this study. Please remember that the BIT values were taken from Sparkes et al. (2015), where the methodology is presented in more detail. However, we can confirm that all GDGTs were analysed on the same system at the same time and can thus be compared. The laboratory at University of Manchester was part of the Schouten et al. study and has implemented protocols that calibrate their GDGT measurements to those reported in the Schouten study. This should allow future researchers to compare between laboratories with more confidence.

62. *L7P11. Can the authors include in their discussion whether the relative abundance of the 3 soil-marker BHPs separately is comparable between ICD settings, but also with marine sediments? Does this support an 'unchanged' terrigenous signal to the marine environment (as stated at L30-35P11).*

> **Response:** Although there are some differences in the relative abundances of the three soil marker BHPs between the different regions (See table below), in all cases BHP **1a** was the most abundant, followed by BHP **1b** and only minor relative amounts of BHP **1b'**. Correlation of concentrations of all 3 pairs of these soil marker BHPs (**1a** vs. **1b**, **1b** vs. **1b'** and **1a** vs. **1b'**) in all ICD samples (from this paper) gives an r2 >0.6 and p value <0.001 for each pair of compounds. This shows that the distribution of the 3 compounds in all ICD samples is comparable across the region. Comparison with the relative abundances of the ESAS sediment samples further show comparable distribution patterns indicating little change in the distribution during transport and deposition in the marine environment.

> We suggest adding the table below to the supplementary information (new Table S4) and the following section to the manuscript to make this clear: "……. Lena region (mean 0.76, range 0.62 – 0.84; Tables 1, S3; Figure 5a). Although there are some differences in the relative abundances of the non-methylated soil markers BHPs between the different regions (Table S4), in all cases BHP **1a** was the most abundant, followed by BHP **1b** and minor relative amounts of BHP **1b'** (See Table S4). Correlation of concentrations of all 3 pairs of these non-methylated soil markers (BHP

1a vs. BHP 1b, BHP 1b vs. BHP 1b' and BHP 1a vs. BHP 1b') in all ICD samples gives an r2 >0.6 and p value <0.001 for each pair of compounds. This shows that the distribution of the non-methylated soil markers in all ICD samples is comparable across the region. Given the higher $R'_{soil}$ values for ICD in the eastern region......."

**Table S4**. Mean relative abundance (+/- 1 standard deviation) of individual non-methylated soil marker BHPs in all Ice complex deposit samples and ESAS sediments from this study.

| Compound | Relative abundance (%) | | |
|---|---|---|---|
| | **1a** | **1b** | **1b'** |
| Terrestrial (ICD) | | | |
| Lena region | $54.7 \pm 11.6$ | $37.7 \pm 10.3$ | $7.6 \pm 3.7$ |
| Indigirka & Kolyma region | $63.9 \pm 5.7$ | $26.8 \pm 5.7$ | $9.3 \pm 5.4$ |
| All | $57.6 \pm 6.1$ | $34.3 \pm 10.4$ | $8.1 \pm 4.3$ |
| ESAS | | | |
| Buor Khaya Bay | $64.0 \pm 4.3$ | $28.0 \pm 3.1$ | $8.0 \pm 3.2$ |
| DLS | $70.4 \pm 1.2$ | $21.4 \pm 1.5$ | $8.3 \pm 1.5$ |
| ISSS nearshore | $58.1 \pm 15.4$ | $32.3 \pm 10.8$ | $9.6 \pm 5.1$ |
| ISSS Offshore | $60.4 \pm 6.3$ | $31.6 \pm 7.1$ | $8.1 \pm 3.3$ |

*63. L22P12. What is the pH range studied in the Hofle et al. (2015) paper? Is this pH range relevant for this manuscript?*

**Response:** The pH range studied in the Höfle et al. (2015) paper was 4.5 to 6.7. Although we do not have pH values for our samples, those from the Höfle et al., study are certainly relevant to this manuscript as all samples were taken from the Lena delta region including samples taken from Kurungnakh Island, in close proximity (<10 km) to the site of the KUR core studied here. This information will be added to the manuscript.

*64.L5P13: I recommend to use the term 'terrigenous', instead of 'terrestrial'.*

**Response:** This will be modified in the revised manuscript

*65. L10P15. The journal name is abbreviated.*

**Response:** This will be modified in the revised manuscript.

*66. L29P16. Vol is mentioned twice*

**Response:** This will be modified in the revised manuscript.

*67.    L33P16. n/a in the reference should be changed to manuscript number.*

**Response:** This will be modified in the revised manuscript.

*68.    L5P18: Journal name is abbreviated*

**Response:** This will be modified in the revised manuscript.

*69.    L9P19: subscript in CO2 and CH4.*

**Response:** This will be modified in the revised manuscript.

70.     *L22P19: replace n/a*

**Response:** This will be modified in the revised manuscript.

71.     *L33P20: replace n/a*

**Response:** This will be modified in the revised manuscript.

72.     *L35P20: If this is a book, please include the publisher.*

**Response:** This will be modified in the revised manuscript.